# Annexin-V stabilizes membrane defects by inducing lipid phase transition

Yi-Chih Lin [1,2], Christophe Chipot [3,4] & Simon Scheuring [1,2]*

Annexins are abundant cytoplasmic proteins, which bind to membranes that expose negatively charged phospholipids in a $Ca^{2+}$-dependent manner. During cell injuries, the entry of extracellular $Ca^{2+}$ activates the annexin membrane-binding ability, subsequently initiating membrane repair processes. However, the mechanistic action of annexins in membrane repair remains largely unknown. Here, we use high-speed atomic force microscopy (HS-AFM), fluorescence recovery after photobleaching (FRAP), confocal laser scanning microscopy (CLSM) and molecular dynamics simulations (MDSs) to analyze how annexin-V (A5) binds to phosphatidylserine (PS)-rich membranes leading to high $Ca^{2+}$-concentrations at membrane, and then to changes in the dynamics and organization of lipids, eventually to a membrane phase transition. A5 self-assembly into lattices further stabilizes and likely structures the membrane into a gel phase. Our findings are compatible with the patch resealing through vesicle fusion mechanism in membrane repair and indicate that A5 retains negatively charged lipids in the inner leaflet in an injured cell.

[1] Department of Anesthesiology, Weill Cornell Medicine, 1300 York Avenue, New York, NY 10065, USA. [2] Department of Physiology and Biophysics, Weill Cornell Medicine, 1300 York Avenue, New York, NY 10065, USA. [3] UMR 7019, Université de Lorraine, Laboratoire International Associé CNRS and University of Illinois at Urbana-Champaign, Vandoeuvre-lès-Nancy F-54500, France. [4] Department of Physics, University of Illinois at Urbana—Champaign, 1110 West Green Street, Urbana, IL 61801, USA. *email: sis2019@med.cornell.edu

**W**hen cells are exposed to excessive mechanical stress, the plasma membrane can locally rupture, resulting in an influx of extracellular $Ca^{2+}$. Uncontrolled and dramatic intracellular $Ca^{2+}$ increase disturbs $Ca^{2+}$-signaling processes[1], and activates $Ca^{2+}$-dependent scramblases[2]. Under these premises, a $Ca^{2+}$-triggered repair system is required to maintain plasma membrane integrity for cell survival and homeostasis in both single-cell and multicellular organisms[3–7]. Other than that, resealing of the plasma membrane may involve intracellular vesicle delivery and $Ca^{2+}$-dependent exocytosis[8], which assuages the significant membrane tension created by the cortical cytoskeleton in cells. The $Ca^{2+}$-triggered repair system also involves cytoskeleton reorganization[9], membrane internalization[10], or shedding of damaged membranes[11]. However, the molecular mechanisms underlying the aforementioned functions are still not understood.

Several proteins have been identified in various eukaryotic cell types to act in membrane repair, such as SNAREs, dysferlin, and annexins[1,12]. Annexins comprise a family of ubiquitous eukaryotic cytoplasmic proteins (~35 kDa; exceptionally ~70 kDa for A6), which bind $Ca^{2+}$ and interact with phospholipids in a $Ca^{2+}$-dependent manner[13,14]. They share a conserved annexin core of four annexin repeats of 70 amino acids, but own variable N-terminal domains, likely related to their distinct physiological functions[15]. The bottom view of the high-resolution crystal structure of A5[16] illustrates the four α-helical annexin repeats in a rhombohedral arrangement, with a pseudo 2-fold axis (Fig. 1a, left), while the side view allows appreciation of the slightly convex membrane-binding face, where $Ca^{2+}$ ions are coordinated by proline-rich loops (Fig. 1a, right). In the presence of a negatively charged bilayer, e.g. the inner leaflet of the plasma membrane, the $Ca^{2+}$ affinity of the A5 membrane-binding face is increased from the mM to tens of μM range[17,18]. As a consequence, A5 buffers excess $Ca^{2+}$ in the cytosol, i.e., concentrations ≥ 10 μM, and is implicated in many functions related to the cell membrane[13,19–21] and its repair[3,12,22].

Among annexins, A5 is known for its ability to self-assemble into highly ordered 2D-lattices on phosphatidylserine (PS)-containing membranes in the presence of $Ca^{2+}$[13]. Electron microscopy and AFM studies have shown that A5 trimers (Fig. 1b) assemble into intricate 2D-lattices[23] (Fig. 1c) at the membrane surface. These A5-lattices were evidenced to form near cell injury sites in response to the dramatic $Ca^{2+}$-influx from the extracellular space[24].

Following A5-recruitment at membrane lesions, it has been proposed that cytoplasmic vesicles could be recruited at the ruptured membrane area and fused with the plasma membrane, thus leading to membrane resealing[1,24,25] (Fig. 1d). Alternatively, the convex membrane-binding face of A5 might provide surface tension along the edges of lesions for a self-resealing process[26] (Fig. 1e). Other members of the annexin family could induce curvature on free-edged membranes, resulting in different morphologies, such as blebs, folds, and rolls[22,27]. However, the structure–function correlation of annexins during membrane repair is still unclear.

HS-AFM offers real-time images of membrane proteins with high spatial (~1 nm lateral and ~0.1 nm vertical) and temporal (~100 ms) resolution under near-physiological conditions. We have employed HS-AFM to visualize that the A5-lattice unit cell comprised two $p6$-trimers and one non-$p6$-trimer, and these biochemically identical trimers possessed different dynamics and $Ca^{2+}$/membrane affinity[18]. Light-induced uncaging of caged-calcium indicated that A5-lattices formed within seconds in response to a $Ca^{2+}$ shock[18]. This molecular-level observation is in agreement with experiments, whereby cells were exposed to laser irradiation, and A5 amassed within seconds near the cell injury

sites[24]. Most recently, we developed HS-AFM height spectroscopy to quantitatively describe A5 diffusion and self-assembly on the membrane in great detail[28]. Other powerful methods to study the dynamics of biomolecules are fluorescence recovery after photobleaching (FRAP), confocal laser-scanning microscopy (CLSM)[29], and molecular dynamics simulations (MDSs)[30]. FRAP is widely used to explore the membrane fluidity[31], while CLSM together with environment-sensitive fluorescent probes can report the lipid order[32]. MDSs offer atomistic details of the biological processes at play.

Here, we combine HS-AFM, FRAP, CLSM, and MDSs to investigate in further detail how A5 membrane binding and self-assembly influence the underlying membrane, shedding light on the mechanism of A5-assisted membrane protection and repair. We provide evidence that A5 anchors negatively charged lipids and induces a phase transition in the underlying lipid bilayer, thereby stabilizing membrane defects for subsequent vesicle-fusion-based membrane resealing.

## Results

**Dynamics and stability of the A5 2D-lattice.** From a kinetic viewpoint, the A5 bulk concentration modulates the adsorption rate, and thus the molecular surface concentration, which in turn leads to a distinct lattice growth rate and surface coverage on the membrane. Therefore, in an attempt to study the dynamics of finite A5-lattices, as would occur in the native cellular system, we adjusted the vesicle-fusion conditions and the A5 bulk concentration to 0.1 μM in order to form small lattices that just partially cover isolated membrane patches (Fig. 2a). HS-AFM movies showed that, under such conditions, A5 coverage can be controlled to ~60% of the membrane surface (Fig. 2b; Supplementary Movie 1). We found that the boundaries of such lattices are highly dynamic, representative of the A5 kinetics at the lattice border in thermodynamic equilibrium. The height difference between the A5 lattice and the membrane is ~2.5 nm, and enables us to designate precisely the lattice border over time. As such, we projected the lattice contours as a function of the border radius ($r$) and angle ($\theta$) into a polar-coordinate system (Fig. 2c). With the benefits of high spatial and temporal resolutions of HS-AFM (0.3 s per frame), the variations of $r$ as a function of the image acquisition time, $t$, at angle $\theta$ reveal association and dissociation events of A5 at the lattice border.

In the radius–contour map as a function of $\theta$ and $t$ (Fig. 2d), the color difference highlights the border variations. During the experimental timespan of 49 s reported here, the lattice expands between $\theta = 60°$ and $120°$, while it shrinks between $\theta = 240°$ and $330°$. In other regions, e.g. $\theta = 150°–210°$, short-scale, fast associations/dissociations occur. As examples, three representative $r$-kymographs at $\theta = 90°$, $180°$, and $270°$ are plotted in Fig. 2e. To obtain kinetic information, we analyzed the dwell times of the state durations before change (association or dissociation) of all $r$-kymographs. The resulting dwell-time histograms fitted by an exponential decay reveal the association ($k_a$) and dissociation ($k_d$) rate constants of A5 molecules at lattice boarders as 2.3 and 2.0 $s^{-1}$, respectively (Fig. 2f). Considering a two-state system, the equilibrium constant ($K_{eq} = k_a/k_d$) of A5 molecules at lattice borders can be estimated as 1.15, suggestive that association of A5 molecules into 2D-lattices is spontaneous and preferred.

We also analyzed the radius difference between two image frames recorded at times $t$ and $t + 0.3$ s to evaluate the occurrence histogram of radius changes on all $r$-kymographs (Fig. 2g, black bars). Given the probability of occurrence, a relative energy diagram of A5 associating to and dissociating from the 2D-lattice can be constructed (Fig. 2g, red profile). This energy landscape has a global energy minimum ($-8.8\ k_BT$) located at $\Delta r = 0$, which

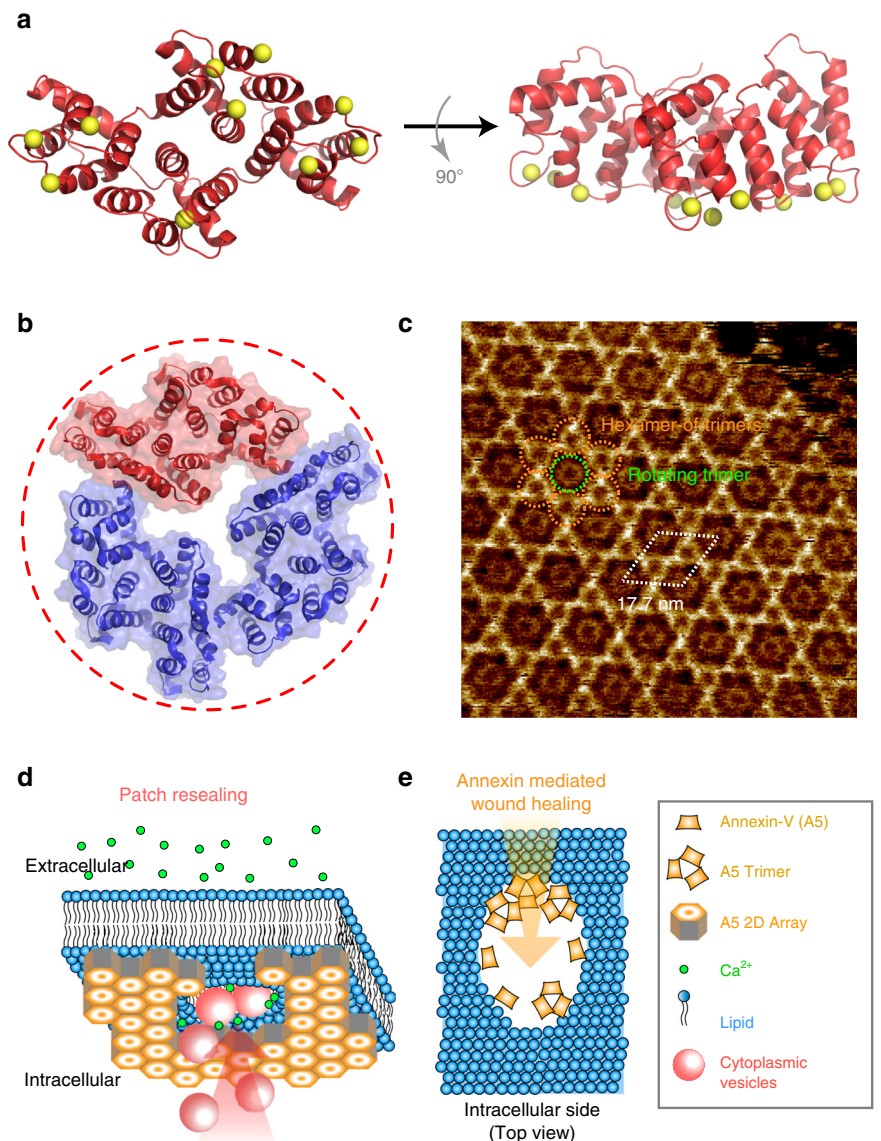

**Fig. 1 Structure, supramolecular assembly, and proposed membrane-repair mechanisms of A5. a** Bottom view (left) and side view (right) of A5 structure (PDB: 1A8A) with bound $Ca^{2+}$ (yellow spheres). The membrane-facing side of the protein is slightly convex (right). **b** Top-view of A5 trimer, one monomer is colored as in **a**. **c** High-resolution HS-AFM image of A5 lattice on a PS-rich membrane. The lattice consists of hexamers-of-trimers (*p*6-trimers) forming a honeycomb pattern; in the center of each honeycomb resides an A5 trimer that displays rotational freedom (non-*p*6-trimer). The unit cell ($a = b = 17.7$ nm, $\gamma = 60°$; white dashed rhomboid) houses 2 *p*6-trimers and 1 non-*p*6-trimer. **d**, **e** Proposed membrane-repair mechanisms: patch resealing (**d**) suggests that the formation of A5 2D-lattices, triggered by the $Ca^{2+}$ influx, stabilizes the membrane near lesions, followed by membrane resealing through fusion of intracellular vesicles. Annexin-mediated resealing (**e**) proposes that A5 assemblies along the edge of lesions provides local surface tension leading to membrane fusion.

represents no radius change. The energy landscape features several rugged regions, where local minima can be found (Fig. 2g, red stars), and the latter are distributed at integral numbers of $\Delta r$, which correspond to the unit of half the lattice length, i.e. the size of an individual A5 trimer. The quasi-symmetric energy landscape indicates a similar probability of A5 binding to and detaching from the lattice. The energy difference between the A5 association/dissociation steps is $\sim 2\,k_B T$, slightly higher than the thermal energy provided. The finite lattice analysis, therefore, demonstrates that A5 2D-lattices are thermodynamically stable, and self-assembly is kinetically favored.

**Annexin-V assembly alters the underlying membrane.** In buffer conditions readily containing 2 mM $Ca^{2+}$, supported lipid bilayer (SLB) are formed on mica via vesicle adsorption, spreading, and

membrane fusion (Fig. 3a; Supplementary Movie 2). Shortening the incubation time of the vesicle sample allowed small membrane patches to be produced, which were diffusive on the mica support, and some eventually fused with each other to form larger patches (Fig. 3b; Supplementary Movie 3). This behavior was expected because the lipids in our minimal membrane system, dioleoylphosphatidylcholine/dioleoylphosphatidyl-serine (DOPC/DOPS) (50:50), have phase-transition temperatures of $-17$ and $-11\ °C$, respectively, and thus form a fluid-phase SLB at room temperature. Subsequent addition of A5 elicited the study of the adsorption and self-assembly of A5 on these membrane patches in real time—this is likely similar for other lipid systems containing negatively charged lipids that interact with A5 over $Ca^{2+}$. Upon addition of A5 into the fluid cell to a final bulk concentration of 0.2 μM, HS-AFM images immediately depicted

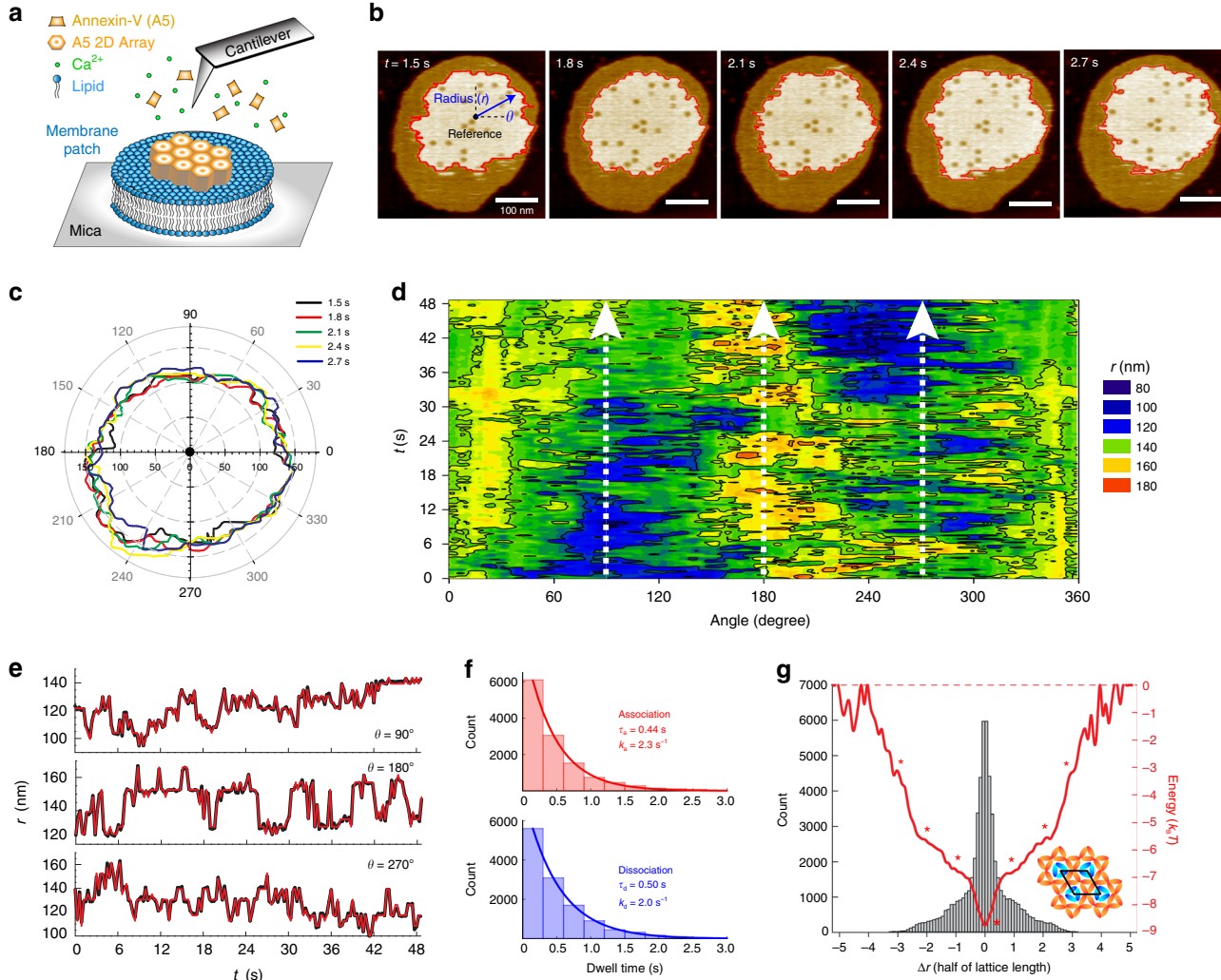

**Fig. 2 A5 association and dissociation kinetics to and from a 2D-lattice. a** Schematic of the experimental setup. Membrane patches (DOPC/DOPS 50:50) are deposited on a mica sample support. The fluid cell contains HEPES buffer (pH 7.4) supplemented with 2 mM $Ca^{2+}$. After addition of A5 to the fluid cell, HS-AFM observes the formation of A5 2D-lattices on membrane patches. **b** HS-AFM image frames (Supplementary Movie 1) showing that the borders of finite A5-lattices are dynamic. HS-AFM movie acquisition speed: 0.3 s per frame. **c** Time-lapse series (lines of different colors) of 2D-lattice boarder in polar coordinates. The reference point is the center of mass of the A5 lattice in the first panel of **b**. **d** Angle($\theta$)–time($t$) map of A5 2D-lattice border radius ($r$). The false color scale represents varying radius from the lattice center to the edge. **e** Kymographs of A5 2D-lattice leading-edge radius at specific angle (white dashed lines in **d**). The red curves represent the idealized traces used for dwell-time analysis. **f** Dwell-time analysis of the association and dissociation kinetics at the leading edges of the A5 2D-lattice. **g** Histogram of the leading-edge radius change. The x-axis is scaled to half of the lattice unit cell length, 8.85 nm, corresponding to the lattice extension/decrease through association/dissociation of one A5 trimer. Each p6 unit cell contains two lattice A5 trimers. The relative energy diagram of A5 2D-lattice association and dissociation (red line) is estimated by $N/N_0 = \exp(\Delta G/k_B T)$, where $N_0$ is the occurred events for $\Delta r = 0$. The zero-energy level is defined at the positions without occurrence.

membrane shrinkage (~40% of the original size), and the rapid adsorption and 2D self-assembly of A5 molecules on the membrane-patch surfaces, which reached full coverage within 80 s. (Fig. 3c; Supplementary Movie 4).

The A5 2D-lattice self-assembly process can be subdivided into five periods. Period 1: molecular A5 adsorption with low surface coverage. Period 2: A5 seeding and self-assembly into small A5 aggregates that can be roughly observed in HS-AFM images. Period 3: coexistence of A5 adsorption and self-assembly with a significant increase in both A5 surface coverage and A5-aggregate number, and the latter one drops off to unity upon formation of one large-scale 2D-lattice. Period 4: growth of the 2D-lattice. Period 5: a single stable 2D-lattice covers the membrane patch that loses its mobility.

We first evaluated the impact of A5 binding and self-assembly on the membrane by plotting the evolution of membrane-patch

size, A5 surface coverage, and the number of A5 arrays as a function of time (Fig. 3d). Visual inspection readily indicated shrinkage of the membrane-patch size during the initial stage of A5 adsorption and diffusion (Fig. 3c, top row). Such shrinkage was never observed in experiments without A5 (Fig. 3b, and see left-hand side of graph in Fig. 3d). While the reason for this shrinkage remains unclear, and lipid shedding by an unknown A5-dependent mechanism cannot be excluded, we propose that membrane seeding fast-moving A5 molecules (invisible in our HS-AFM frames, because their diffusion speed is too high compared with the image acquisition) readily coalesce lipids by means of their $Ca^{2+}$-liganded face, hence leading to an overall shrinkage of the membrane surface area.

Quantitatively, the membrane area continuously decreases during A5 adsorption, in line with the interpretation that A5 on the membrane coalesce lipids, until the size of the A5 aggregates

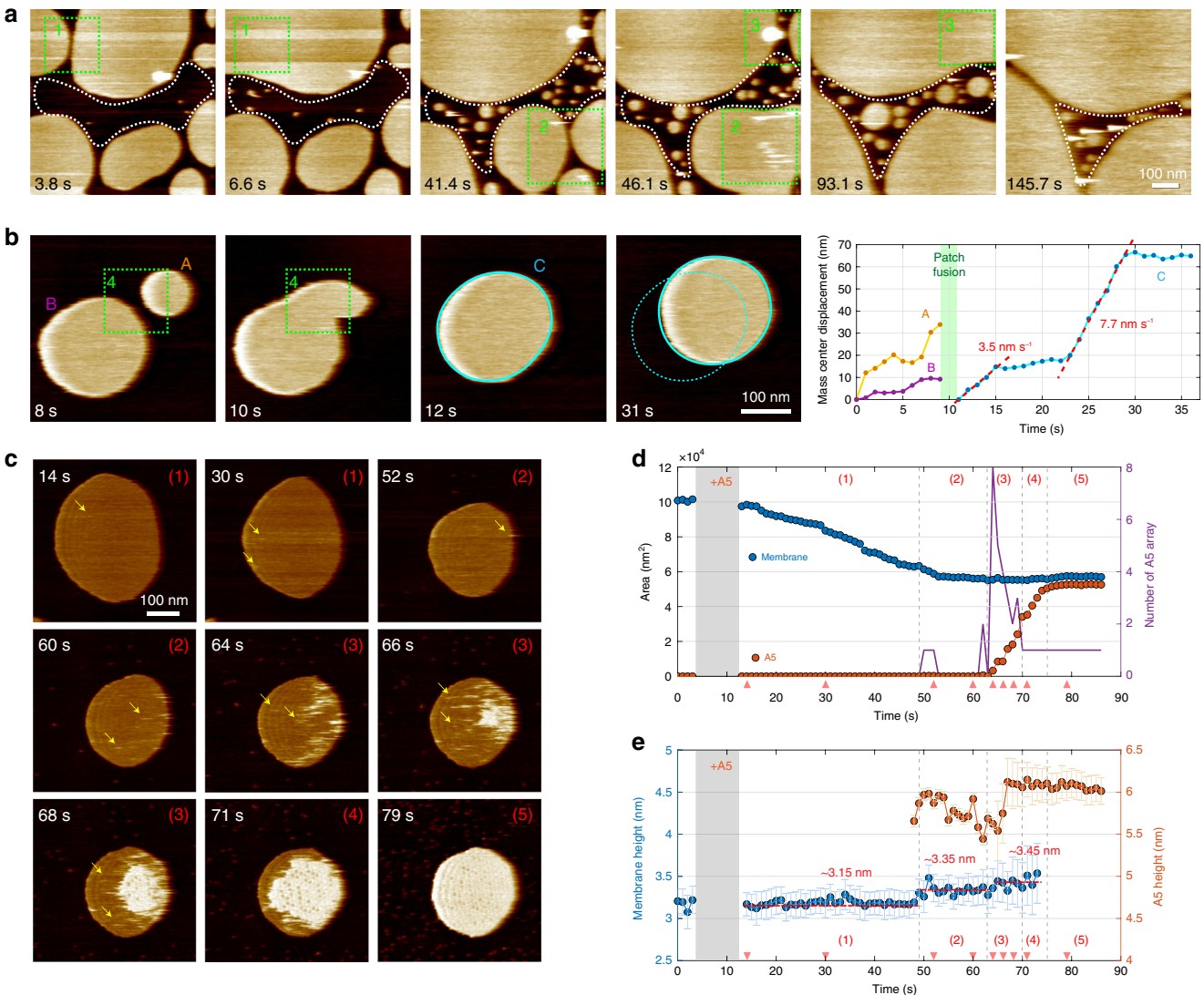

**Fig. 3 A5 2D-lattice growth alters the characteristics of the underlying lipid membrane. a** HS-AFM image frames (Supplementary Movie 2) showing the efficient formation of a supported lipid bilayer (SLB) membrane on mica through vesicle adsorption and spreading (white dashed outlined) and patch fusion (green dashed rectangles). **b** In the absence of additional vesicles deposited from buffer, the fusion of two small membrane patches can produce a new larger membrane patch that has freedom in surface diffusivity (Supplementary Movie 3). Right graph panel: movement estimated by mass center displacement through fusion process and surface diffusion. **c** HS-AFM image sequence from a longer HS-AFM movie (acquisition speed: 1 s per frame, Supplementary Movie 4) showing membrane-patch size decrease during A5 adsorption, self-assembly, and 2D-lattice growth. Yellow arrows indicate fast-diffusive, small A5 aggregates. **d** Time-lapse analysis of area and the number of A5 arrays (minimum size $\geq 200$ nm$^2$). Red arrows indicate the time of representative HS-AFM frames shown in **c**. Based on the changes in membrane area, A5 coverage, and A5-aggregate number, five distinct periods during the A5 self-assembly process are identified as following: (1) membrane shrinkage upon A5 addition, (2) A5 self-assembly into small A5 aggregates, (3) abundant A5 adsorption and coalescence of A5 aggregates in a large-scale 2D-lattice, (4) growth of a 2D-lattice until the membrane was fully covered, and (5) a solid A5-protective membrane. The assigned periods in **d** are labeled in the top-right corner of corresponding images in **c**. **e** Time-lapse analysis of mean height above the mica plane of the membrane patch and A5 aggregates. Each data point is the difference between centers Gaussian fits of the height distributions of the mica, the membrane, and the A5 with the error bar determined by the peak width of the Gaussian fit of the membrane/A5 height distribution (Supplementary Fig. 1). The thickness of bare membrane and A5-covered membrane are 3.15 ± 0.2 nm and 6.1 ± 0.2 nm, respectively. The A5 self-assembly process promotes an average membrane thickness increase to 3.35 ± 0.2 nm and 3.45 ± 0.3 nm during period (2) and (3), respectively. Detailed comparison among height histograms is provided in Supplementary Fig. 1.

increases and the A5-protected membrane is constant in size (Fig. 3c, middle and bottom row panels, Fig. 3d, blue and orange traces). During this period, A5 adsorption is rapid and self-assembly into a complete lattice is achieved in a short time of ~20 s. This finding suggests that membrane shrinkage is mediated upon membrane binding of A5 molecules, whereas the larger A5 2D-lattices protect membranes. Noteworthily, the membrane patch remains immobile during the entire A5 self-assembly process, and ever afterwards.

Next, we examined the topography of the membrane and the A5 lattice (Figs. 3c, e). Indeed, one of the strengths of AFM due to its sensitivity in the z-dimension is measuring the membrane thickness that reflects the lipid-membrane phase state[33–36]. Depending on the lipid types, and acyl-chain length and saturation, differences in thickness ranging from 0.2 nm to 1.0 nm were typically found between liquid and gel phases. Analyzing the height distribution histograms as a function of the A5 self-assembly process, we found that the mean thickness of the

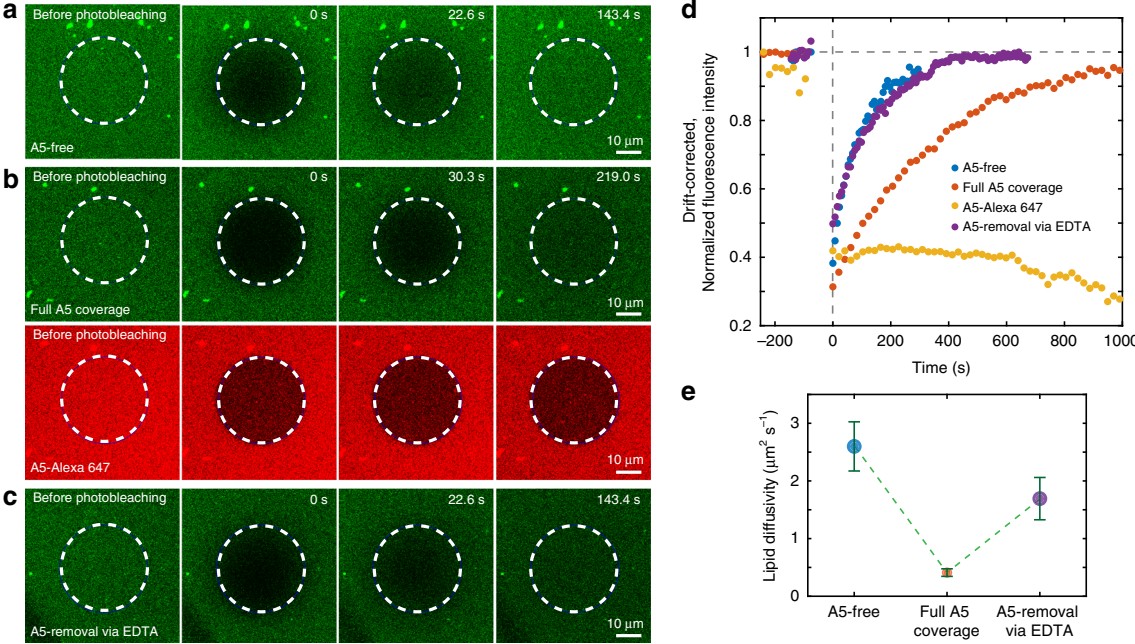

**Fig. 4 A5 2D-lattice modulates the diffusivity of underlying lipids. a–c** Consecutive fluorescence recovery after photobleaching (FRAP) experiments on SLB membrane (DOPC/DOPS/NBD-PC 19:20:1) on glass measured in the conditions of **a** A5-free (SLB only), **b** full A5 coverage, and **c** A5-removal upon EDTA-addition. Two separate spectral channels with different excitation lasers were used to monitor NBD-PC and A5-Alexa 647 (mixture of A5:A5-Alexa 647 at a ratio of 2:1), respectively. The white dashed circles indicate the photobleaching area. **d** Time-lapse analysis of FRAP experiments within the photobleaching areas shown in **a–c**. The presence of the A5 2D-lattice slows down the fluorescence recovery of the underlying membrane (orange). The A5 2D-lattice is stable and shows no fluorescence recovery. **e** Calculated lipid diffusivity in the subsequent FRAP experiments. The formation of A5 2D-lattice decreases the mobility of underlying lipids, which was recovered by removing A5 molecules through EDTA-addition. Each data represent the mean ± s.d. (mean ± standard deviation) of $n \geq 10$ FRAP areas.

underlying membrane, determined by Gaussian fits, increased by ~0.2 nm and then ~0.3 nm during periods 2 and 3 (Fig. 3e, blue points), respectively. Such an increase in height, therefore, suggests that A5 association and assembly lead to a change in the lipid organization toward a more ordered state.

We further estimated another remarkable, albeit hitherto underappreciated way whereby A5 membrane adsorption and A5-lattice formation influence the underlying bilayer: A5 forces the membrane to accept a very high local $Ca^{2+}$ concentration in its vicinity. Depending on the experimental conditions, A5 binds between 2 and 11 $Ca^{2+}$ ions on its membrane-facing side according to the various high-resolution structures available (on average, 5.3 $Ca^{2+}$-ions, Supplementary Note 1). Thus, given the number of A5 in the lattice unit cell (three trimers), the size of the unit cell (271.3 $nm^2$), and the confinement of $Ca^{2+}$-ions into a volume of ~1 nm thickness above the bilayer, a local $Ca^{2+}$-concentration between 110 mM and 606 mM is predicted. Such high $Ca^{2+}$-concentrations are known to alter bilayer properties[37], as we will further study below.

**Annexin-V assembly alters the diffusion of underlying lipids.** To provide further details about the changes of the membrane characteristics induced by A5 self-assembly, we first performed FRAP experiments[31] to study the lipid diffusivity of fluorescence-labeled SLB (doping the DOPC/DOPS 1:1 mixture with 2.5% NBD-PC). FRAP events were monitored on SLBs alone (Fig. 4a), SLBs with adsorbed A5 (Fig. 4b, two channels), and on SLBs where A5 was removed again following addition of EDTA (Fig. 4c). Comparing the FRAP curves recorded under these different conditions, the A5-covered SLB (Fig. 4d, orange) displayed a much slower recovery rate compared with SLBs without

A5 (Fig. 4d, blue) and SLBs after removal of A5 (Fig. 4d, purple). Concurrently, with the SLB, the fluorescence intensity of the A5 2D-lattice (Fig. 4b, red channel; Fig. 4d, yellow) was monitored, and exhibited, as expected, no recovery— reflecting the rigid and stable architecture of A5 2D-lattices without molecular rearrangements. Using the half time of recovery and the effective bleach radius for a circular bleached region[38], the lipid diffusivity was determined as $2.6 \pm 0.4$, $0.4 \pm 0.1$, and $1.7 \pm 0.4$ $\mu m^2 s^{-1}$ for the SLB alone, SLB covered by A5, and the SLB after A5-removal, respectively (Fig. 4e). Thus, the presence of an A5 lattice slows down lipid diffusion to only ~15% of the original diffusivity. EDTA-addition successfully removed A5 from the SLB (Supplementary Fig. 2), and restored lipid diffusivity to ~65% of the A5-free condition. Considering the tripartite interplay of A5, $Ca^{2+}$-ions and negatively charged lipids, the NBD-PC head group should be negligibly attracted by $Ca^{2+}$-A5 compared with DOPS. Therefore, the reduction of lateral diffusion calculated from the NBD-PC FRAP experiments should be representative for the bilayer and indicative of an ordered membrane phase.

**Annexin-V assembly alters the order of underlying lipids.** Aside lipid diffusivity, a parameter directly related to membrane phase transition is the lipid order. This property can be explored using environment-sensitive dyes that change their fluorescence intensity or color in response to molecular order. For example, the membrane-incorporated dye, Laurdan, displays a ~50 nm blue-shifted emission in a liquid-ordered compared with a liquid-disordered phase membrane[32,39,40]. Thus, the ratiometric measurement of the fluorescence intensity recorded in two separate spectral channels, known as the generalized polarization (GP) value, can quantitatively assess the lipid order[40]. Here, we also

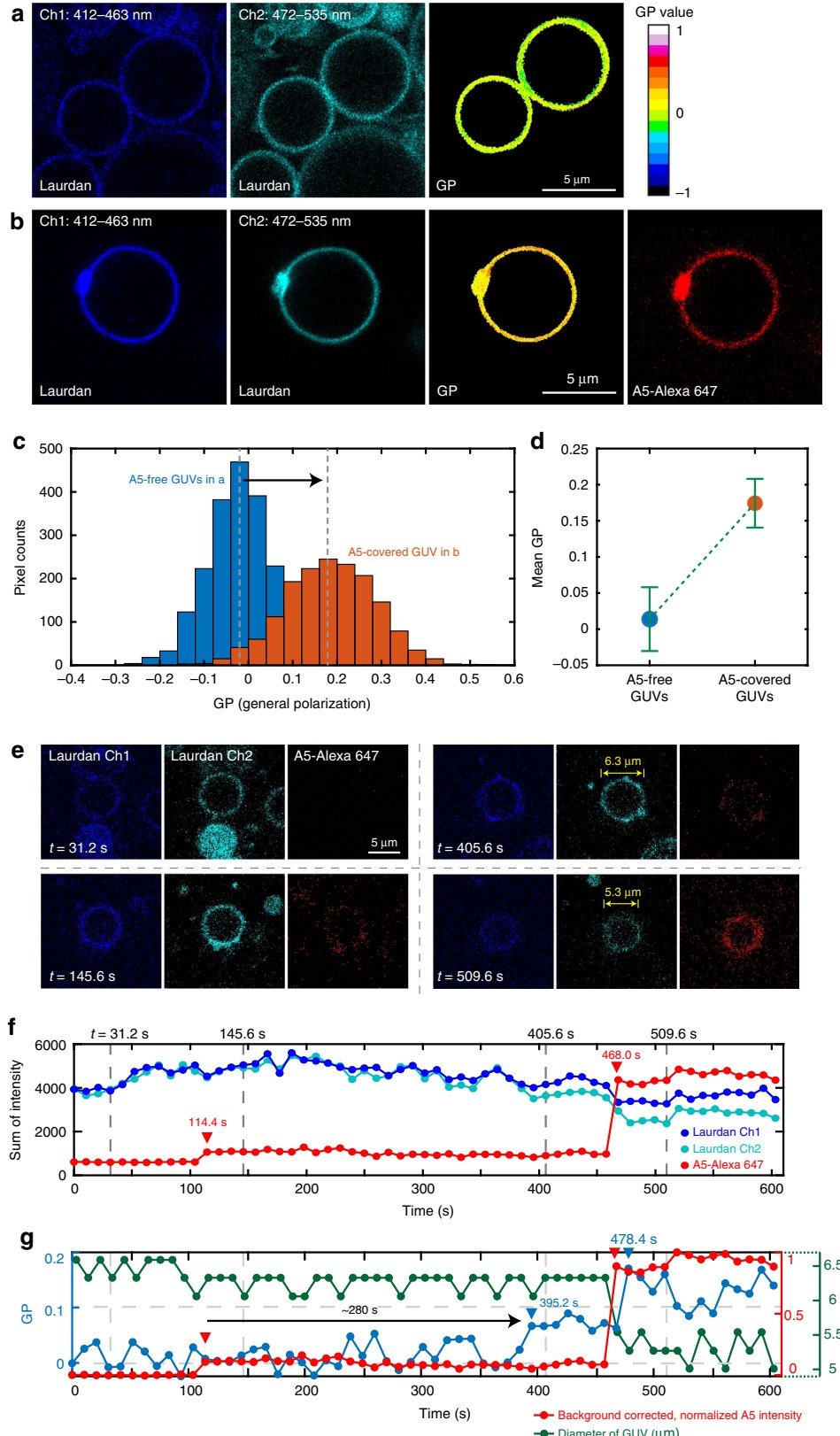

utilized Laurdan to investigate the lipid order of giant unilamellar vesicles (GUVs), of the same lipid composition as the SLBs used in HS-AFM experiments, before, during, and after A5 addition. The usage of GUVs instead of SLBs was because of the relative orientation of the membrane-incorporated Laurdan electronic transition moment with respect to the polarization plane of the excitation laser in CLSM[39]. Thus, we employed CLSM focusing at the equatorial plane of free-standing GUVs doped with Laurdan

**Fig. 5 A5 small aggregates and 2D-lattice can mediate the order of underlying membrane. a, b** CLSM fluorescence images of GUVs (DOPC/DOPS 50:50, focusing at the equatorial plane) staining with environment-sensitive dye, Laurdan, recorded in the conditions of **a** A5-free or **b** full A5 2D-lattice coverage, respectively. Three spectral channels were used to monitor the fluorescence of Laurdan (Ch1: 412–463 nm; Ch2: 472–535 nm) or A5-Alexa 647, respectively. Based on the Equation 1, the generalized polarization (GP) image of GUVs can be determined using the averaged fluorescence images of Ch1 and Ch2 ($n = 20$). **c** GP Histograms of two A5-free GUVs (blue) and a GUV fully covered by A5 (orange) as shown in **a, b**, respectively. The presence of A5 2D-lattice shifts the most probable peak in GP histogram toward to a higher value (black arrow), which specifies the membrane becomes more ordered. **d** The mean GP value of the A5-free GUVs (blue, 0.01 ± 0.04) or A5-covered GUVs (orange, 0.17 ± 0.03), respectively. Each data represent for mean ± s.d. among different GUVs with $n \geq 10$. **e, f** Representative CLSM fluorescence image frames (**e**) and time-dependent fluorescence intensity traces (**f**) showing the impacts of A5 2D-lattice self-assembly on a GUV, following the subsequent additions of A5 at 114.4 s and 468.0 s, respectively (indicated by red triangles in **f**). **g** Time-lapse analysis of GP value (blue curve), background corrected, normalized A5 intensity (red curve), and diameter of the GUV (green curve). This GP value is calculated using the sum of fluorescence intensity in two spectral channels of Laurdan. Although the first A5 addition makes GUV partially covered by A5 with ~10% coverage, a small increasement in GP value can still be observed at ~395 s (blue triangle, GP = 0.07, ~280 s after A5 addition). The subsequent second A5 addition increases the GP value up to 0.169 at 592.8 s, which is highly compatible with the mean GP value observed for A5-covered GUVs in **d**. Upon two A5 additions, the diameter of the GUV decreases especially during the second A5 addition, along with simultaneous GP value increases. Statistical analysis on the shrinkage of GUVs diameter is provided in Supplementary Fig. 3.

(minimizing the photoselection effect[39]), and recorded two separate spectral channels in A5-free (Fig. 5a) and A5-full-coverage (Fig. 5b) conditions. The GP values of the GUVs (third panel in Fig. 5a, b) were calculated following[40]:

$$GP = \frac{I_{(ch1)} - GI_{(ch2)}}{I_{(ch1)} + GI_{(ch2)}}, \tag{1}$$

where $I$ is the fluorescence intensity acquired in the indicated spectral channel. $G$ is the calibration gain factor set to ~1/3 for our confocal setup, so that $\langle GP \rangle$ is close to 0 in the reference conditions (A5-free). Upon addition of A5 and full coverage of the GUV (Fig. 5b), the GP histograms shift and the GP peak value changes from $-0.02$ to $0.18$ (Fig. 5c). Statistically, $\langle GP \rangle$ value obtained from different GUVs ($n \geq 10$) shows the same increase from $0.01 \pm 0.04$ to $0.17 \pm 0.03$ in the presence of A5 (Fig. 5d). Thus, the increase of $\langle GP \rangle$ value demonstrates that adsorption and coverage with A5 brings the underlying lipids into a more ordered state.

We also monitored the fluorescence of Laurdan and A5 on a free-standing GUV with the fastest CLSM speed (10.4 s per frame recording three channels) upon injection of A5 into the sample solution (Fig. 5e, f). The time-dependent A5 fluorescence intensity documents the two subsequent A5 injections at 114.4 s and 468.0 s (Fig. 5f, red arrowheads), which resulted in a ~10%- and full A5 coverage on the GUV, respectively (Fig. 5f, g, red trace). The first A5 injection led to only ~10% of the full saturation, and only flickering fluorescence intensities at specific regions were detected, likely representative of small, mobile A5 aggregates on the GUV. The calculated GP values (Fig. 5g, blue trace) before the first A5 injection (A5-free) and after the second A5 injection (fully A5-covered) are compatible with the mean GP values measured in the static experiments (Fig. 5d). Furthermore, we observed that, following the first A5 addition, a significant increase of the GP value from ~0 to ~0.07 occurred over ~280 s, and plateaued after 395.2 s. This observation combined with the A5 fluorescence channel provides further evidence that small, mobile A5 aggregates on GUVs readily modulate the lipid order. At low A5 surface coverage, the time course between the HS-AFM and the CLSM experiments are somewhat different, because (i) the A5-assembly conditions are different, i.e. continuous in HS-AFM and stepwise in CLSM and (ii) the surface area of membrane patches (~0.1 μm², Fig. 3c) and GUVs (~110 μm²) are very different.

Moreover, akin to the HS-AFM experiments, we observed shrinkage of the GUV diameter from ~6.3 to ~5.3 μm after the second A5 addition (Fig. 5e, middle panels at $t = 405.6$ s and 509.6 s). The time-dependent analysis of the GUV diameter (Fig. 5g, green curve) further indicates compaction and changes

in the lipid order upon A5 membrane binding and 2D-lattice formation. Although the GUV diameter should not be considered an accurate value (because GUV equatorial section should change when the diameter varies), it evidences a correlation between lipid order and the shrinkage of the GUV diameter (Supplementary Fig. 3).

Put together, we have found five independent observables in three experimental approaches characterizing A5-dependent membrane-patch changes, i.e., surface area shrinkage (by HS-AFM of SLBs and CLSM of GUVs), loss of mobility (by HS-AFM of SLBs), decreased diffusivity (by FRAP of SLBs), increased thickness (by HS-AFM of SLBs), and increased lipid order (CLSM of Laurdan in GUVs), all pointing to the same interpretation: the tripartite interplay of A5, $Ca^{2+}$-ions and lipids modulates the order of the membrane underneath A5 and induces a transition from liquid-crystal to gel phase (see the Discussion section). In other words, A5 self-assembly orders A5 protein units, and in turn the A5-bound $Ca^{2+}$-ions, propagating the order into the underlying lipid bilayer.

**Atomistic view of the A5/$Ca^{2+}$/lipid interaction.** To gain atomistic insight into structural changes of lipids underlying the A5/$Ca^{2+}$ 2D-lattice, we turned to MDSs (Fig. 6a). As a preamble to production simulations, the membrane was thermalized over a brief period of 10 ns (assay 1, 0.05 M $Ca^{2+}$), during which the simulation cell underwent rapid contraction (Fig. 6b, left). In the subsequent 2.9 μs of the simulation, the surface area of the membrane patch decreased further by ~7% (Fig. 6b, dark gray). While shrinking of a lipid bilayer is directly visible—in stark contrast with the reference assay bereft of $Ca^{2+}$-ions, in which x, y-dimensions plateau (assay 4, inset in Fig. 6b)—the spatial reorganization of the acyl chains is slow and precludes observation of a spontaneous liquid-crystal-to-gel-phase transition over the typical timescales of brute-force MDS. To address this limitation, in an approach akin to that followed by Melcrová et al.[37], a large amount of $Ca^{2+}$ ions was added to the ones restrained at the crystallographic positions in A5. Within the next 1.2 μs, the surface area shrank by another 4% (assay 2, Fig. 6b, medium gray). Increase of the $Ca^{2+}$ concentration from 0.48 to 0.54 M led to an additional decrease by 3% over 3 μs (assay 3, Fig. 6b, light gray), i.e. an overall reduction of the surface area of the thermalized and equilibrated membrane patch by ~15%. While complete crystal-to-gel-phase transition goes beyond the scope of the present simulations, retraction of the DOPC/DOPS lipid bilayer in the x, y-directions and expansion in the z-direction are suggestive of stretching and partial ordering of the acyl chains, which prefaces isomerization to an all-trans conformation. Lipid order parameters, $S_{CD}$, determined as $\langle P_2(\cos \alpha) \rangle$, where $P_2$ is the second Legendre polynomial and α is the angle between a C–H bond and the bilayer

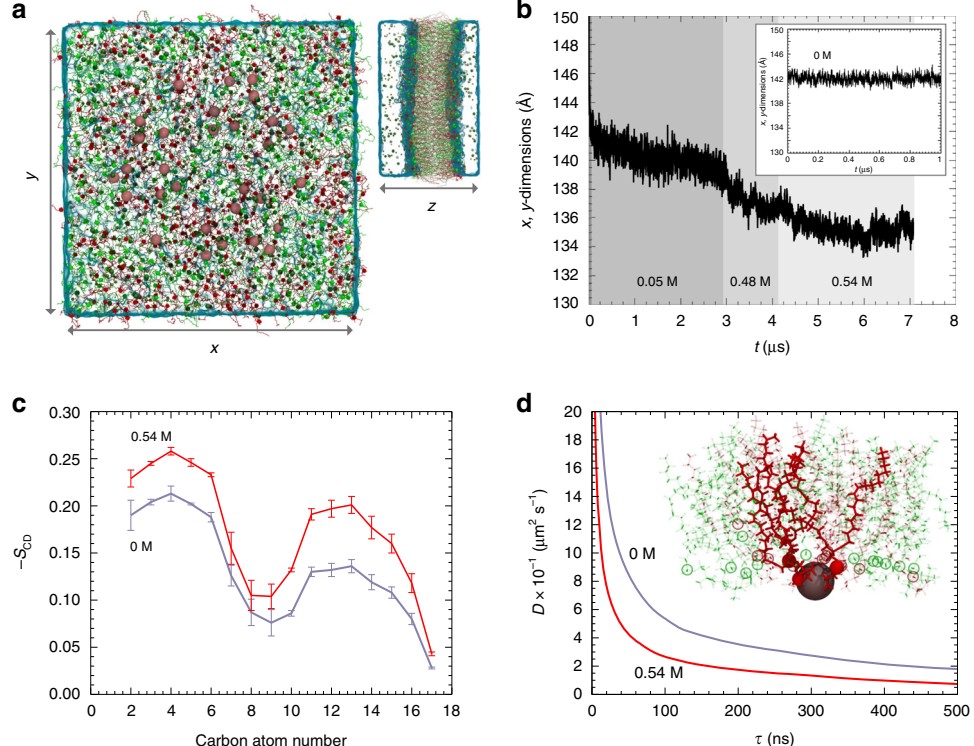

**Fig. 6 Molecular dynamics simulation (MDS) of a DOPC/DOPS bilayer exposed to a Ca$^{2+}$-lattice, extreme Ca$^{2+}$-concentrations and in absence of Ca$^{2+}$.** **a** Top and side views of the computational assay. DOPC and DOPS lipids are shown in green and red, respectively. Calcium, sodium, and chloride are shown as pink, yellow, and blue van der Waals spheres. **b** Time evolution of the $x$, $y$-dimensions of the simulation cell at increasing calcium concentration. Reference assay devoid of calcium (inset). **c** Order parameter of the DOPC lipids in the reference assay devoid of calcium (lavender) and at high calcium concentration (red), determined as averages over sn-1 and sn-2 chains, in a time frame of 1 μs. Each data represent for mean ± s.d. of the order parameter. **d** Diffusivity of DOPS lipids measured from their mean-squared displacement over a period of 1 μs. Typical configuration of a calcium ion chelated by DOPS (inset).

$z$-direction, confirm that ordering of the acyl chains increases at 0.54 M Ca$^{2+}$-concentration (Fig. 6c, red line), as compared with the lipids in the absence of Ca$^{2+}$ (Fig. 6c, lavender line). At the same time, chelation of the cations by the head groups reduces lipid mobility within the membrane. Estimation of DOPS lateral diffusion based on the mean-squared displacement of the phosphorus atoms at $\tau = 500$ ns yields a diffusivity of 17.9 μm$^2$ s$^{-1}$ in the absence of Ca$^{2+}$-ions, dropping to 7.3 μm$^2$ s$^{-1}$ at 0.54 M (Fig. 6d), in agreement with the previous experimental measurements at 0 and 0.15 M of 12.4 and 8.3 μm$^2$ s$^{-1}$, respectively[41]. The theoretical estimates of the diffusivity ought to be considered, however, with caution, in view of the non-linear growth of the mean-squared displacement of the phosphorus atoms as a function of time, a hallmark of incomplete sampling and possible subdiffusion[42]. Although MDSs provide a higher diffusivity compared with our FRAP experiments (likely due to the use of SLB in FRAP experiments vs free-standing bilayers in MDSs), both data give the same interpretation, that A5 2D-lattices slow down the diffusion of underlying lipids.

**The role of A5 molecules in Ca$^{2+}$-triggered membrane repair.** We have shown that both the thermodynamic stability and kinetics favor self-assembly of the A5 2D-lattice on bilayers containing negatively charged lipids (Fig. 2) and a transition to the gel phase in the membrane induced by A5 self-assembly (Figs. 3–6). Thus, changing our current view, we have to consider the A5 2D-lattice, the Ca$^{2+}$-ions and the membrane as one physical ensemble. In order to gain insight into how this system is involved in membrane protection and repair, we next set out to study the A5–Ca$^{2+}$-lattice–membrane interaction with other membranes.

Toward this end, we supplemented A5-protected membrane patches (Fig. 7a, patches A and B) with additional DOPC/DOPS vesicles into the HS-AFM fluid cell and observed the subsequent membrane formation on the mica that readily contained the membranes covered with A5 2D-lattices (Supplementary Movie 5). The newly formed membranes lack A5-lattices, and thus are fluid and diffusive. As a result, these vesicles fuse with each other to almost complete surface coverage (Fig. 7a, $t = 0.0$ s to $t = 53.2$ s). While fusion of the newly added vesicles is efficient, the A5-covered membranes seem to be initially excluded from the fusion process, and the first membrane contact with patch A is formed at $t = 29.6$ s (yellow dashed circle) and at $t = 41.6$ s for patch B (magenta dashed circle). It is noteworthy that ~95% of the circumference of patch A is surrounded by a gap to the otherwise complete surrounding bilayer ($t = 41.6$ s), indicative of the incompatibility of mixing two different phases of the membrane underlying the A5 lattice (gel phase) and the newly formed empty bilayers (fluid phase). Moreover, lipid diffusion into the gel-phase patch seems slow, which is manifested by remnant A5-lattice integrity from initial contact with the fluid phase from $t = 29.6$ s to $t = 53.2$ s (patch A), after which the A5 lattice gradually dissociates into small, diffusive A5 aggregates (Fig. 7a, stripes indicated by green arrows in $t = 104.0$ s). The final membrane morphology reveals several thin gaps left between the newly formed membrane and the previously A5-covered membrane. The process can be quantitatively followed by plotting the surface area of A5 patches A and B, the formation of the new membrane increasing surface coverage from ~30 to ~97% within

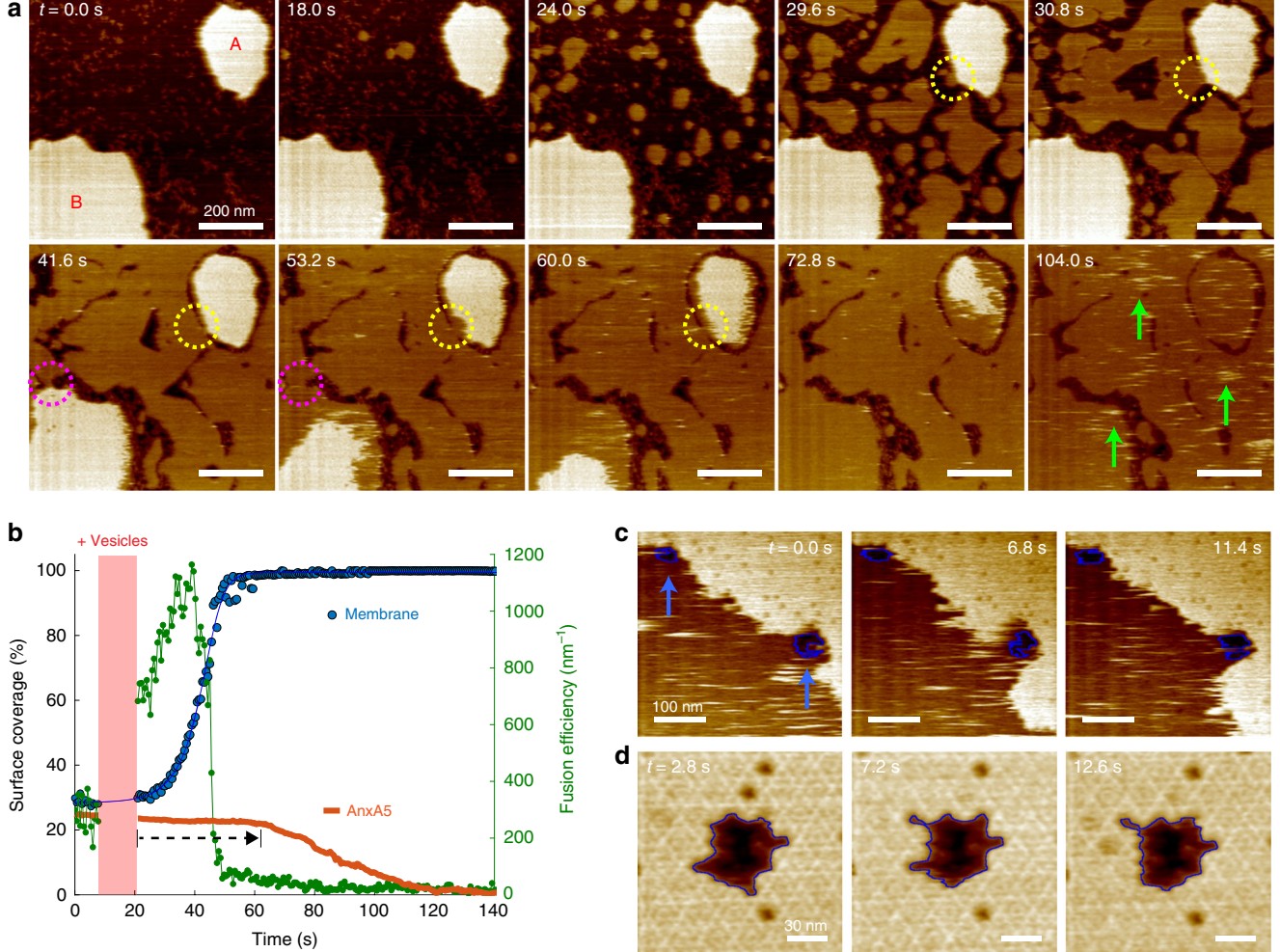

**Fig. 7 HS-AFM of A5-assisted membrane healing processes. a** HS-AFM image frames (Supplementary Movie 5) showing the formation of membrane, following the addition of small unilamellar vesicles into fluid cell, in the presence of preformed membrane patches covered with A5 2D-lattices. The colored circles indicate membrane fusion locations between A5-protected membranes and newly formed membranes. Green arrows in frame 104.0 s indicate topographical stripes resulting from highly mobile A5 after membrane fusion. **b** Graph of the time-lapse evolution of the integrity of A5 patches A and B, total membrane surface coverage, and membrane fusion efficiency, estimated by the ratio of circumference and membrane area on newly formed membrane. A5-protected membranes remain robust beyond the time of first successful fusion events. **c** A5 dynamics (blue arrows) (Supplementary Movie 6) and **d** A5 2D-lattice dynamics (blue lines) (Supplementary Movie 7) near membrane defects following the addition of supplemental A5 to the HS-AFM fluid cell.

the scanning area, and the fusion efficiency of new membranes, as a function of time: The A5 membrane patches A and B remain basically unaltered (Fig. 7b, black dashed arrow), while new protein-free membrane patches are efficiently formed (Fig. 7b, green) leading to almost total surface coverage.

In another attempt, we examined annexin-mediated membrane healing near small lipid-membrane defects through injection of additional A5 into the HS-AFM fluid cell. After monitoring distinct areas, both surface-mobile A5 (Fig. 7c; Supplementary Movie 6) and A5 2D-lattices (Fig. 7d; Supplementary Movie 7) were unable to actively reseal even small membrane defects. A5 proteins rearranged at lattice borders or moved around the perimeter of small defects, without ever bridging over the defect. Lattice formation, therefore, needs the underlying bilayer.

Our HS-AFM observations altogether suggest that A5-assisted membrane healing occurs through membrane stabilization by dint of A5 2D-lattice formation, and that the A5-protected membrane first hinders further wound expansion. A5 remains stable until a fusion event with a lipid vesicle is successful (Fig. 7a,

b). In contrast, A5 cannot achieve membrane resealing, or bridging of membrane defects independently (Fig. 7c, d).

## Discussion

Harnessing the high spatial and temporal resolutions of HS-AFM, we observe directly the impact of A5 self-assembly on PS-rich membranes in the presence of $Ca^{2+}$. A5 molecules bind peripherally, self-assemble on the membrane surface, and modulate the membrane properties, namely thickness and diffusivity, indicative of a phase transition from fluid to a more ordered phase. FRAP and CLSM experiments further provide quantitative assessments of the impact of A5 self-assembly on membrane properties, including the decrease of diffusivity of the lipids and the increase of their acyl-chain order. According to crystallographic data, each A5 monomer carries between 2 and 11 (in average 5.3) $Ca^{2+}$-ions on its membrane-binding face (Supplementary Note 1), acting as a "rigid" molecular glue between the protein units and the negatively charged lipids. Under these premises, MDSs demonstrate that addition of $Ca^{2+}$-ions lead, indeed, to membrane

ordering, thickening, and shrinkage, consistent with a liquid-to-gel phase transition. We, therefore, conclude that the initial two steps of A5-mediated membrane repair, involve the recruitment of A5 and $Ca^{2+}$-ions to the membrane, followed by A5-guided self-assembly of both into a 2D-lattice. HS-AFM, FRAP, CLSM, and MDS all show that this order is translated to the underlying membrane, leading to thickening and stiffening in relation to decrease of diffusivity and increase of acyl-chain order. These changes in the physicochemical properties of the bilayer have likely been designed to protect the membrane around defects and prevent further expansion of defects. This design principle might be of importance in a cellular context, where membrane tension along the edge of a wound would lead to enlargement of the defect if the sturdy A5 lattice would not surround and protect the lesion through anchoring the negatively charged lipids in place.

Addition of A5 units to the A5-protected membranes with defects never led the lattice to grow over these defects and/or reseal them. In contrast, we found that supplementing the system with vesicles (mimicking intracellular vesicles) could lead to fusion of the latter with A5-protected membrane edges. Following this event, the A5 lattice remains intact over a considerable amount of time, likely providing and maintaining stability until the membrane fusion and healing process is completed.

In summary, we expect that the key components of the A5-mediated membrane-repair machinery should, at least, include A5 and intracellular vesicles. The peripheral binding and self-assembly of A5 enable quick membrane stabilization, while preventing further expansion and propagation of defects. Concomitantly, delivery of intracellular vesicles will reseal the membrane defects. Other physiological processes, such as exocytosis[43] or release of caveolae[44], may evidently contribute positively to overcome the membrane tension induced by the cortical cytoskeleton during membrane repair, a process that deserves further study in its own right.

The exposure of PS on the outer membrane leaflet is an apoptotic signal in cells[45]. This will happen in cells that suffered from membrane lesion through two paths: First, PS will flip leaflets at the membrane injury site. Second, the inflow of extracellular $Ca^{2+}$ will activate the $Ca^{2+}$-dependent lipid scramblases TMEM16, abolishing the native membrane asymmetry[46,47]. Thus, by binding and buffering $Ca^{2+}$ [48] and, hence, silencing scramblases during the initial moments of a membrane lesion, and by anchoring PS in the inner leaflet, thereby rendering PS inaccessible to scramblases, annexin provides two essential functions to safeguard the cells from entering apoptosis.

In this report, we have provided additional insights into A5-assisted membrane healing. We show that an essential step is the anchoring of PS and an order transfer from the protein lattice to the membrane, inducing a phase transition. This process seems at first glance thermodynamically unfavorable, as it corresponds to cooling down the lipids. The energetic cost is, however, likely balanced by the network formation of the lipid, the $Ca^{2+}$-ions and the protein, which engage into favorable noncovalent bonds and change hydration interfaces. Many protein systems involved in membrane trafficking and remodeling, like clathrin, dynamin, caveolin, and ESCRT-III, also engage in repetitive patterns through self-assembly on membranes. Modulation of the physicochemical properties of the underlying membrane induced by such protein systems might, therefore, constitute a far more common mechanism than hitherto appreciated.

## Methods

**Sample preparation.** The A5 used in this study was purchased from Sigma-Aldrich (Annexin-V, 33kD from human placenta). All lipids (dioleoylphosphatidyl-choline (DOPC), dioleoylphosphatidyl-serine (DOPS), and 1-myristoyl-

2-{6-[(7-nitro-2-1,3-benzoxadiazol-4-yl)amino]hexanoyl}-sn-glycero-3-phosphocholine (NBD-PC)) were purchased from Avanti polar lipids. The fluorescent dyes, including the Alexa Fluor 647 dye (A5-Alexa 647) for A5-labeling and Laurdan, were purchased from Thermo Fisher Scientific.

To prepare small unilamellar vesicles (SUVs), lipids were first dissolved in chloroform at a ratio of DOPC:DOPS = 1:1 or DOPC:DOPS:NBD-PC = 19:20:1 for HS-AFM or FRAP experiments, respectively. The solvent solubilized mixed lipids were dried by a nitrogen flow and for further drying kept in a vacuum chamber for 12 h. Then the dried lipid was resuspended into the buffer solution containing 20 mM HEPES NaCl at pH 7.4, 150 mM NaCl, and 2 mM $CaCl_2$. As the final step of the lipid preparation, the suspension was sonicated for 30 min to obtain SUVs. Before HS-AFM imaging, 1.5 μl of the diluted SUV solution with a total lipid concentration of 0.1 mg ml$^{-1}$ was deposited onto freshly cleaved mica <1 minute to form membrane patches through vesicle fusion, and then rinsed with the same buffer. Using HS-AFM, the process of bilayer formation could directly be observed. A5 2D-crystals were grown on supported lipid bilayers by addition of A5 to preformed membrane patches. For FRAP experiments, the 50 μl of the SUV solution with a total lipid concentration of 1 mg ml$^{-1}$ was deposited onto a glass cover slip for 10 min incubation to form a SLB, and then rinsed with the same buffer. The A5 additions are a mixture of A5 and A5-Alexa 647 at a ratio of 2:1 from stock solutions.

The giant unilamellar vesicles (GUVs) were prepared using the electroformation method[49]. Lipids were dissolved in chloroform at a ratio of DOPC:DOPS = 1:1, homogeneously spread onto two clean ITO glasses and let dry, followed by further drying in a vacuum chamber for 12 h. An electroformation chamber assembled by the two ITO glasses and an O-ring was filled up with 200 mM sucrose solution and sealed with silicone to avoid evaporation. The chamber was subjected to a sinusoidal wave with amplitude 1.6 V at 10 Hz for 2 h, and then to a square wave of 1.6 V at 10 Hz for 15 min to produce GUVs. The GUVs were suspended in the buffer solution, and then stained with Laurdan (5 μM final concentration) before CLSM experiments. In total, 100 μl of the stained GUV solution was gently placed onto cover glass for CLSM fluorescence imaging. To monitor the impact of the A5 self-assembly on the GUVs, total 20 μl (10 μl per addition) of the fluorescent A5 mixture was injected into the solution phase.

**HS-AFM.** All movies in this study were taken by amplitude modulation mode HS-AFM (RIBM, Japan). Short cantilevers (NanoWorld, Switzerland) with spring constant of 0.15|Nm$^{-1}$, resonance frequency of 0.6 MHz, and a quality factor of ~1.5 in buffer, were used. The tip-sample interaction was minimized to avoid scanning interference, typically the ratio of the set-point and free amplitude was ~0.9 with a free amplitude of ~1.0 nm. HS-AFM experiments were successfully performed in replicates for more than five times on different samples, days and HS-AFM tips.

**HS-AFM image processing and data analysis.** The HS-AFM movies were drift corrected and contrast adjusted by a laboratory build image analysis software in ImageJ. The kinetics analysis of A5 lattice boarder was performed by a Leading Edge ImageJ plugin[50] and self-written routines in MATLAB R2018a. The membrane mass center, membrane area, height histogram of HS-AFM frame, and fusion efficiency were analyzed using ImageJ. The Gaussian fits to the height histogram were analyzed by self-written routines in MATLAB R2018a.

**Fluorescence recovery after photobleaching (FRAP).** FRAP experiments were performed with a confocal laser-scanning microscope ZEISS LSM 880 (Carl Zeiss AG, Oberkochen, Germany) equipped with an Airyscan detection unit and a high sensitivity GAsP detector for visible detection. To maximize the resolution enhancement, we used a high numerical aperture (NA) oil immersion objective (Plan-Apochromat 40×/1.3 Oil DIC M27; Zeiss). Two spectral channels were used to record the fluorescence of NBD-PC lipid (laser excitation: 458 nm; emission: 470–600 nm) and A5-Alexa 647 (laser excitation: 633 nm; emission: 640–750 nm). The photobleaching was achieved using a 405 -nm laser operated at maximum laser power. FRAP experiments were successfully performed in replicates for more than three times on different samples and days.

**FRAP image processing and data analysis.** For FRAP experiments, the fluorescence intensity within the photobleaching area was first calibrated by another control area without photobleaching, and then normalized to the mean of fluorescence intensity measured before photobleaching. Based on a simplified equation proposed by Kang et al.[38], the lipid diffusivity can be extracted from the confocal FRAP data by:

$$D = (r_n^2 + r_e^2)/8\tau_{1/2} \qquad (2)$$

where $D$ (μm$^2$ s$^{-1}$) is the diffusion coefficient and $\tau_{1/2}$ (s) is the half time of recover. $r_n$ (μm) and $r_e$ (μm) are the radii of the designed circular bleach region and the effective bleach radius, respectively. Thus, we separately analyzed $\tau_{1/2}$ from FRAP curves and $r_e$ from the fluorescence images, and then calculated the lipid diffusivity using Equation 2.

**CLSM of Laurdan-doped GUVs**. CLSM fluorescence imaging spectrum was performed with a confocal laser-scanning microscope ZEISS LSM 880 (Carl Zeiss AG, Oberkochen, Germany) equipped with an Airyscan detection unit and a high sensitivity GAsP detector for visible detection. To maximize the resolution enhancement, we used a high numerical aperture (NA) oil immersion objective (Plan-Apochromat 63×/1.4 Oil DIC M27; Zeiss). All imaging was performed using Immersol 518 F immersion media (Carl Zeiss). Laser gain, detector gain, and pixel dwell times were adjusted to maintain the lowest laser power and highest signal to noise ratio in order to avoid saturation and bleaching effects. Three spectral channels were used to record the fluorescence of Laurdan (laser excitation: 405 nm; channel 1: 412–463 nm; channel 2: 472–535 nm) and A5-Alexa 647 (laser excitation: 633 nm; emission: 640–750 nm). CLSM experiments were performed in replicates for more than three times on different samples and days.

**CLSM image processing and data analysis**. As we described in the article, the GP value can be calculated from the GUV fluorescence images using the ratiometric measurement of the Laurdan fluorescence intensity recorded in two separate spectral channels (channels 1 and 2) based on Equation 1[40]. The calibration gain factor ($G$) for our confocal setup was determined using the ratiometric value between channel 1 and channel 2 measured in A5-free conditions, which is ~1/3. In such conditions, the lipids in the GUVs are in the liquid-disordered state represented by the mean GP value close to 0. The time-dependent analysis of the GUV diameter is performed by the 360-fold average[51] after aligning the GUV's circular center positions to the image center.

**MD simulations**. The computational assay consisted of a lipid bilayer formed by 320 1,2-dioleylphosphatidylserine (DOPS) units and 320 1,2-dioleylphosphatidylcholine (DOPC) in equilibrium with 35,244 water molecules, corresponding to an initial cell dimension of about $150 \times 150 \times 85$ Å$^3$. The solution was buffered with NaCl at a concentration of about 150 mM. In total, 31 Ca$^{2+}$ ions were added to the surface of the membrane at their crystallographic location in the three-dimensional structure of A5 (Fig. 6a), corresponding to a concentration of about 0.05 M (assay 1). The position of the calcium ions was restrained by means of a soft harmonic potential with a force constant of 1 kcal mol$^{-1}$ Å$^2$. To probe the effect of the Ca$^{2+}$ interfacial concentration on the phase transition of the lipid bilayer, Ca$^{2+}$ ions were subsequently added to assay 1, resulting in a number of cations equal to 295 and 335, which corresponds, respectively, to a concentration of 0.48 M (assay 2) and 0.54 M (assay 3). A fourth, reference assay, devoid of calcium ions, was built for comparison purposes (assay 4).

All the MDSs reported herein were performed employing the parallel, scalable program NAMD 2.12[52]. Periodic boundary conditions (PBCs) were applied in the three directions of Cartesian space. Water was described by the TIP3P model[53], and the lipid bilayer by the all-atom CHARMM36 force field[54,55]. A mass repartitioning scheme was introduced, allowing the equations of motion to be integrated with a time step of 4 fs, using the r-RESPA multiple time-step algorithm[56]. Covalent bonds involving hydrogen atoms were constrained to their equilibrium length by means of the RATTLE[57] and SETTLE[58] algorithms. The temperature and the pressure were maintained at 298 K and 1 atm, respectively, using Langevin dynamics and the Langevin piston method[59]. Long-range electrostatic forces were taken into account by means of the particle mesh Ewald algorithm[60]. A 12-Å cutoff was applied to truncate van der Waals and short-range Coulombic interactions. Visualization and analyses of the MDS trajectories were performed with the VMD program[61].

Simulation of the liquid-crystal-to-gel-phase transition of a lipid bilayer is notoriously challenging and remains unamenable to unbiased, brute-force atomistic MDSs, owing to the μs-to-ms timescales spanned by the isomerization and collective reorganization of the acyl chains[62]. While coarse-grained approaches have proven successful to address cooperative phase transition at a semiquantitative level[63], they also lack the fine atomic detail required to capture the chelation of the Ca$^{2+}$ ions by the lipid head groups. One might speculate that brute-force MDSs could be employed fruitfully to simulate the reverse, gel-to-liquid-crystal transition by decreasing the Ca$^{2+}$ concentration in lieu of increasing the temperature, as is done traditionally to induce acyl-chain disorder[64]. However, unlike temperature, which affects all degrees of freedom of the lipids through energy equipartition, removal of Ca$^{2+}$ ions is unlikely to accelerate gel-to-liquid-crystal transition by promoting isomerization across the entire acyl chain. It might be argued that the incomplete sampling of the collective degrees of freedom of the membrane in brute-force MD, highlighted in the non-diffusive behavior of the lipids, can be magnified by force-field inaccuracies. In particular, additivity of the potential energy function and absence of charge transfer phenomena, responsible for the suboptimal description of lipid–ions interactions, have prompted the design of a mean-field ansatz, whereby the charge borne by the participating ions is scaled down[37], an approach that we have used in this work.

**Reporting Summary**. Further information on research design is available in the Nature Research Reporting Summary linked to this Article.

## Data availability

Data supporting the findings of this manuscript are available from the corresponding author upon reasonable request. A Reporting Summary for this Article is available as a Supplementary Information file. The source data underlying Fig. 2c–g, 3b, d, e, 4d, e, 5c, d, f, g, 6b–c, 7b and Supplementary Figs. 1–3 are provided as a Source Data file.

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

## Acknowledgements
This research was supported by the grant NIH NCCIH DP1AT010874 (to S.S.). We thank the WCM imaging facility (Optical Microscopy Core) for support with FRAP and CLSM experiments.

## Author contributions
Y.L. and S.S. designed the study. Y.L. performed HS-AFM, FRAP, and CLSM experiments and data analysis. S.S. supervised HS-AFM, FRAP, and CLSM experiments and data analysis. C.C. performed and analyzed MDS experiments. Y.L., C.C. and S.S. wrote the paper.

## Competing interests
The authors declare no competing interests.
