## [Peer Review File · Nature Communications]

Reviewers' comments:

Reviewer #1 (Remarks to the Author):

Using solid-supported model membranes (SLBs) consisting of a 1:1 mixture of DOPC and DOPS, Lin and coworkers analyzed the association of annexin A5 (A5) with this model membrane by high speed atomic force microscopy (HS-AFM). A5 had previously been shown to form two-dimensional protein lattices on PS-containing membranes and this property had been linked to a function in plasma membrane repair, a complex Ca²⁺-driven process that involves membrane resealing and cytoskeleton rearrangements. Lin et al now confirm by HS-AFM that A5 readily forms 2D lattices on SLBs which are highly dynamic at their boundaries. Furthermore, they show that the A5-membrane association is accompanied by a decrease of the area of the respective solid-supported SLB patch, most likely reflecting a stiffening and thickening of the bilayer, ie a liquid-to-gel phase transition, following A5 binding. The authors also attempted to simulate this phase transition by MD simulations using high Ca²⁺ concentrations as a surrogate for the Ca²⁺-dependently bound A5. Finally, again using HS-AFM Lin et al analyzed the formation of new membrane patches on solid supports that contained preformed membrane areas covered by A5 lattices. This experiment revealed that new membrane patches, which formed after the addition and fusion of SUVs on the solid support, fused preferentially with one another and not with the A5-covered membrane areas. Based on these data the authors developed a model for the role of A5 in plasma membrane wound repair whereby Ca²⁺ elevation first triggers the formation of an A5 lattice close to the wound and this lattice then transfers order to the underlying membrane preventing further expansion of the membrane defect. Intracellular vesicles can concomitantly fuse with the plasma membrane at A5-free areas thereby resealing the membrane defect.

This paper uses an elegant approach to dynamically study membrane-protein interactions at high resolution. It reports interesting results, however the novelty is at present still limited. 2D-A5 lattices on model membranes and their involvement in plasma membrane repair as an emergency response preventing expansion of the membrane defect had been reported before. Moreover, previous publications even made use of A5 mutants defective in lattice formation to show that this property is directly linked to a function in membrane repair. The important novel result of the Lin et al manuscript concerns the change in membrane order that is apparently induced by the A5 lattice but this evidence is currently still indirect. More experiments are required to support this conclusion (see below).

Specific points:

1. Fig 1. Is the image in Fig 1c taken from a previous publication? If so, this paper should be cited. The model in Fig 1e is difficult to understand. What should the green circles represent, Ca²⁺ ions or a different lipid species?
2. Page 4, first paragraph. The authors changed the bulk A5 concentrations used in their assays. Do these concentrations reflect A5 concentrations occurring in cells?
3. The model membrane chosen (DOPC:DOPS at 1:1) is rather artificial. Are the main A5-induced changes (membrane stiffening) also observed when lipid compositions more closely reflecting a cellular membrane are used (eg membranes containing PE, PA, PIs and cholesterol)?
4. Fig. 3 c,d. The decrease in the size of the membrane patch that is triggered by A5 addition occurs before the actual formation of an A5 lattice (Fig 3c and d). The authors refer to this time point as period 1, molecular A5-adsorption with an extremely low surface coverage. How can this low surface coverage induce a phase transition in the entire membrane area, ie the postulated stiffening and thickening? I could believe that this lipid phase transition is triggered by a relatively rigid 2D-protein/Ca²⁺ lattice but by not by highly dynamic A5 molecules or trimers at very low coverage.
5. To support their conclusion the authors should employ other methods showing more directly that a lipid phase transition occurs upon A5 binding, eg calorimetry or fluorescence polarization of membrane-incorporated dyes.
5. Fig. 3 e. The height of the membrane measured here (3.15 nm) is rather low for a typical phospholipid bilayer. Is this due to the method or phospholipid composition used?
6. The MD simulations showed effects at rather high Ca²⁺ concentrations (appr 0.5 M). Would these Ca²⁺ concentrations induce phase transitions by themselves (ie in the absence of A5) when employed in the AFM setup?

Reviewer #2 (Remarks to the Author):

The manuscript authored by Lin et al. tried to elucidate the molecular mechanism of membrane wound-healing by annexin A5. The authors first utilizes high-speed AFM to visualize the dynamics of annexin A5 on the flat lipid bilayer. They found that A5 assembled into 2D-lattice and stabilized the lipid bilayer. Then the authors used molecular dynamics simulation and demonstrated that a local high concentration of Ca²⁺ induced an ordered state of lipids. The experiments were well-designed and well-conducted. Data were presented properly and seem highly intriguing in terms of the function of A5 itself. On the other hand, only a weak evidence has been shown on the "phase transition" and "stabilization" of lipids by A5, which the authors claimed is the new finding in this study. Especially, the interpretation of Figure 3, which is the most important figures in the manuscript, seems unclear to reach the conclusion that A5 stabilizes the lipid bilayer by inducing a phase transition. They repeatedly claimed that the formation of 2D-lattice of A5 induced a phase transition of lipids and restricted the mobility. However, their HS-AFM observation demonstrated that the height of the lipid bilayer had changed before A5 formed the 2D-lattice. In addition, there is no direct evidence that the lipid phase changed from liquid crystal to gel states. Since A5 strongly binds to lipids in a Ca²⁺-dependent manner, the lateral mobility of the lipids in the bilayer is restricted by 2D-lattice formation of A5. However, this is just a "restriction" of lipid movement by A5, but not a "phase transition" of lipids. The followings are the comments related to the issues mentioned above.

Major comments:

1. In Figure 3c and d, the authors demonstrated that the addition of A5 to the lipid bilayer shrank a patch of lipid bilayer on a flat substrate. Based on the time-course observation of the membrane area and A5 binding, they found 5 distinct periods in the A5 assembly process. Immediately after the addition of A5, a patch of lipid bilayer started to shrink (Figure 3c), resulting in a slow decrease of lipid area (Figure 3d). The authors claim that the shrinkage is caused by lipid-bound A5 molecules, although they are invisible on the AFM images due to their fast 2D-diffusion. This might be a possible interpretation, but other interpretations might also be possible, and even more feasible. For example, free A5 in the chamber deprived lipids from the edge of the 2D-patch, or just accelerated the dissociation of lipids from the 2D-patch by unknown mechanisms. Therefore, the authors have to show a direct evidence of membrane "shrinkage", to distinguish from a reduction of lipid molecules from the 2D-lipid bilayer.
2. Following the membrane shrinkage in Period (1), the authors claimed that the lipid-bound A5 became visible as a small seeding/aggregate (Period (2)). However, it is very hard to see A5 molecules in Figure 3c and Movie S4. The authors should show the image and indicate the small patch of A5 by an arrow.
3. In Figure 3e, the height of the lipid bilayer and that of A5 lattice from the mica surface during the A5 assembly was measured. At 49 seconds, when Period (1) ended and Period (2) started, the height of the membrane suddenly increased from 3.15 nm to 3.35 nm by 0.2 nm. In addition, the height of A5 was measured between 5.5 and 6.0 nm. However, based on Figure 3d, there is no A5 2D-lattice on the lipid bilayer during Period (2), although a small aggregate emerged in the beginning of Period (2) and immediately disappeared after several seconds. So, I wonder how they measured the height of A5 in the absence of A5 2D-lattice on the lipid bilayer. Does it mean the height of diffusive A5 on the lipid bilayer? If so, the authors should show the A5 height plot in Period (1) as well. It is strange that Figure 3d does not contain the A5 height in Period (1).
4. Related to the previous comment, the authors claimed that the sudden change in the height of the lipid bilayer in Period (2) was due to phase transition of lipids from liquid crystal to solid states. However, in this period (Period (2)), most of A5 molecules existed on the surface of the lipid bilayer in a dispersed and diffusive state (line 179-181), and had not formed 2D-lattice yet. This means that diffusive A5 caused a phase transition of the lipids. I wonder how this happened. The author should explain how phase transition had been caused by diffusive A5.
5. In addition, the height of the lipid bilayer further increased in Period (3) by 0.1 nm. However, the authors did not explain how this happened. Is this another phase transition of lipids? The authors should clarify this.

6. The phase transition occurred after the shrinkage of the lipid patch (Period (1)) stopped. It was not clearly stated what is the relationship between the membrane shrinkage and the phase transition of the lipids. The authors should clarify the relationship among the following issues: membrane shrinkage by diffusive A5 (Period (1)), phase transition of the lipids by diffusive A5 (Period (2)), further phase transition of the lipid (?) and 2D-assembly of A5 lattice (Period (3)).

7. In Figure 4, the authors performed MD simulation to mimic the membrane shrinkage and ordered state of lipids in the presence of high concentration of Ca^{2+} . They claimed that lipid bilayer of DOPC/DOPS was shrunk in x-y direction and transformed into more ordered state in a Ca^{2+} -dependent manner. They estimated a local concentration of Ca^{2+} induced by 2D-lattice of A5, and performed the simulation. However, based on Figure 3, the membrane shrinkage occurred in the absence of A5 2D-lattice (Period (1)), and stopped during the lattice formation (Periods (2) and (3)). Therefore, if they wanted to simulate the membrane shrinkage, they had to estimate local concentration of Ca^{2+} in Period (1), where A5 is dispersed and diffusive, and not visible by AFM.

Minor comments

Line 135: Figure 2f should be Figure 2g?

Legend of Figure 3: period (ii) and (iii) should be period (2) and (3)?

Responses to Reviewer's Comments

Reviewer's comments: Black

Author's Response: Blue

Responses to Reviewer's Comments

We thank the reviewers for their thorough and well detailed criticism. Responding to the reviewers' comments clearly made this work much better. It was particularly 'positive' that both reviewers' critiques identified the same weaknesses in our former manuscript version, and thus made it valuable for us to engage into extensive additional experiments to answer their concerns. We have performed two novel and independent sets of additional experiments, namely fluorescence recovery after photobleaching (FRAP) of SLBs and confocal laser scanning microscopy (CLSM) of GUVs supplemented with an environment-sensitive fluorescence dye, and present these results in the new sections of "Annexin-V assembly alters the diffusion of underlying lipids" and "Annexin-V assembly alters the order of underlying lipids" (pages 8 to 10) and new Figures 4 and 5. Using FRAP, we show that lipid diffusivity is decreased to ~15% of the original diffusivity underneath A5-lattices. Using CLSM, we show the order change of the lipids during and after A5-adsorption on membranes (GUVs of the same lipid composition).

Thus, along with the original HS-AFM and the MDS experiments, we now present five independent observables by four independent experimental approaches characterizing A5-dependent membrane patch changes, *i.e.* surface area shrinkage (by HS-AFM of SLBs and CLSM of GUVs and MDS), loss of bilayer mobility (by HS-AFM of SLBs), decreased diffusivity (by FRAP of SLBs and MDS), increased thickness (by HS-AFM of SLBs and MDS), and increased lipid order (CLSM of Laurdan in GUVs and MDS).

We hope that the reviewers find our revised manuscript and our responses convincing.

Responses to Reviewer's Specific Comments

Reviewer #1 (Remarks to the Author):

Using solid-supported model membranes (SLBs) consisting of a 1:1 mixture of DOPC and DOPS, Lin and coworkers analyzed the association of annexin A5 (A5) with this model membrane by high speed atomic force microscopy (HS-AFM). A5 had previously been shown to form two-dimensional protein lattices on PS-containing membranes and this property had been linked to a function in plasma membrane repair, a complex Ca²⁺-driven process that involves membrane resealing and cytoskeleton rearrangements. Lin et al now confirm by HS-AFM that A5 readily forms 2D lattices on SLBs which are highly dynamic at their boundaries. Furthermore, they show that the A5-membrane association is accompanied by a decrease of the area of the respective solid-supported SLB patch, most likely reflecting a stiffening and thickening of the bilayer, *ie* a liquid-to-gel phase transition, following A5 binding. The authors also attempted to simulate this phase transition by MD simulations using high Ca²⁺ concentrations as a surrogate for the Ca²⁺-dependently bound A5. Finally, again using HS-AFM Lin et al analyzed the formation of new membrane patches on solid supports that contained preformed membrane areas covered by A5 lattices. This experiment revealed that new membrane patches, which formed after the addition and fusion of SUVs on the solid support, fused preferentially with one another and not with the A5-covered membrane areas. Based on these data the authors developed a model for the role of A5 in plasma membrane wound repair whereby Ca²⁺ elevation first triggers the formation of an A5 lattice close to the wound and this lattice then transfers order to the underlying membrane preventing further expansion of the membrane defect. Intracellular vesicles can concomitantly fuse with the plasma membrane at A5-free areas thereby resealing the membrane defect.

This paper uses an elegant approach to dynamically study membrane-protein interactions at high resolution. It reports interesting results; however the novelty is at present still limited. 2D-A5 lattices on model membranes and their involvement in plasma membrane repair as an emergency response preventing expansion of the membrane defect had been reported before. Moreover, previous publications even made use of A5 mutants defective in lattice formation to show that this property is directly linked to a function in membrane repair. The important novel result of the Lin et al manuscript concerns the change in membrane order that is apparently induced by the A5 lattice but this evidence is currently still indirect. More experiments are required to support this conclusion (see below).

Specific points:

1. Fig 1. Is the image in Fig 1c taken from a previous publication? If so, this paper should be cited. The model in Fig 1e is difficult to understand. What should the green circles represent, Ca²⁺ ions or a different lipid species?

Author Reply: The image in Figure 1c is novel unpublished data. The green circles in Figure 1f represent Ca²⁺ ions. To avoid confusion, we listed the symbol representations in the right of Figure 1f.

2. Page 4, first paragraph. The authors changed the bulk A5 concentrations used in their assays. Do these concentrations reflect A5 concentrations occurring in cells?

Author Reply: Annexins are known to be highly abundant cytoplasmic proteins. However, we were unable to find a quantitative information about the A5 concentration in the cytosol. In previous literature (Bouter, A. et al. Nat Commun, 2011, doi:10.1038/ncomms1270) it has been shown that A5 successfully self-assembled at membrane injury site which indicates the A5 quantity in cell is high enough to form A5 2D-lattices on membrane. However, in cells, upon membrane injury, A5 lattices are not infinite, as they are in typical in vitro experiments. Therefore, in this work, we changed the bulk A5 concentration in order to control A5 surface concentration for self-assembly to an amount that lattice borders can be observed.

3. The model membrane chosen (DOPC:DOPS at 1:1) is rather artificial. Are the main A5-induced changes (membrane stiffening) also observed when lipid compositions more closely reflecting a cellular membrane are used (eg membranes containing PE, PA, PIs and cholesterol)?

Author Reply: The reviewer raises a good point. Indeed, the membrane properties should be somewhat changed when the lipid composition is more complex. For example, the presence of cholesterol can increase membrane stiffness. We have chosen a minimal system, where both lipids have T_m far below experimental conditions. Importantly, the tripartite interactions between A5, Ca²⁺-ions, and negatively charged lipids (PS) are the major players involved in the A5-dependent membrane repair process, and we show how this tripartite interaction anchors the negatively charged lipid in the membrane – this is most likely also the case of instead of the PS another negatively charged lipid was interacting with the A5 over Ca²⁺.

4. Fig. 3 c,d. The decrease in the size of the membrane patch that is triggered by A5 addition occurs before the actual formation of an A5 lattice (Fig 3c and d). The authors refer to this time point as period 1, molecular A5-adsorption with an extremely low surface coverage. How can this low surface coverage induce a phase transition in the entire membrane area, ie the postulated stiffening and thickening? I could believe that this lipid phase transition is triggered by a relatively rigid 2D-protein/Ca²⁺ lattice but by not by highly dynamic A5 molecules or trimers at very low coverage.

Author Reply: To address this (major) reviewer's comment, we performed additional experiments to study the lipid properties and order upon A5 addition. Using the membrane-incorporated dye Laurdan,

which reports about order changes of the lipid chains via a ratiometric fluorescence changes, we found the membrane order of GUVs (of the same lipid composition as HS-AFM measurement) shift to higher values, corresponding to a phase change, after the formation of A5 2D lattices (new Figure 5a-5d, page 10). At low A5 surface coverage, where small and mobile A5 aggregates can be detected in the confocal fluorescence images, the lipid order of GUV readily gradually increased to higher values (new Figure 5e-5g, page 10). Therefore, we expect that the “highly dynamic” A5 molecules readily influence the order of the underlying lipids – we have to remind that (i) even isolated A5 diffuse almost an order of magnitude slower than lipids in a bilayer (Heath G. R. & Scheuring S., Nat Commun, 2018, DOI: 10.1038/s41467-018-07512-3), and (ii) each A5 trimer ‘binds’ between 9 and 30 negatively charged lipids, and thus locally readily creates order in the bilayer (without forming a 2D lattice).

5. To support their conclusion the authors should employ other methods showing more directly that a lipid phase transition occurs upon A5 binding, eg calorimetry or fluorescence polarization of membrane-incorporated dyes.

Author Reply: We thank the reviewer for the valuable suggestion. To support our conclusions, we performed additional FRAP and CLSM experiments and present results in the new sections of “Annexin-V assembly alters the diffusion of underlying lipids” and “Annexin-V assembly alters the order of underlying lipids” (pages 8 to 10) and new Figures 4 and 5.

6. Fig. 3 e. The height of the membrane measured here (3.15 nm) is rather low for a typical phospholipid bilayer. Is this due to the method or phospholipid composition used?

Author Reply: The bilayer thickness depends of course on the lipid compositions. Based on our experience, the membrane thickness of 3.15nm is normal for a SLB composed of the highly fluid DOPC:DOPS 1:1 mixture, which has been examined at different days and using different HS-AFM scanners. Literature values are comparable.

7. The MD simulations showed effects at rather high Ca²⁺ concentrations (appr 0.5 M). Would these Ca²⁺ concentrations induce phase transitions by themselves (ie in the absence of A5) when employed in the AFM setup?

Author Reply: Our MD simulations first use Ca²⁺ ions restrained at the crystallographic positions of the A5 2D lattice in order to characterize how the A5 lattice order is translated to A5-bound Ca²⁺-ions, and then imposed onto the lipid bilayer. These simulations showed that the negatively charged lipids go anchored on these immobilized Ca²⁺. However, computational MD simulation work at timescales much too short to see changes in lipid order. Therefore, we used additional high Ca²⁺ concentrations. This pointed into the direction of the induction of lipid phase transition. However, although high Ca²⁺ concentration could potentially induce a phase transition, the lack of physiological significance of high Ca²⁺ concentration in solution and geometric constrains cannot provide further biological insight for the A5-membrane interactions.

Reviewer #2 (Remarks to the Author):

The manuscript authored by Lin et al. tried to elucidate the molecular mechanism of membrane wound-healing by annexin A5. The authors first utilize high-speed AFM to visualize the dynamics of annexin A5 on the flat lipid bilayer. They found that A5 assembled into 2D-lattice and stabilized the lipid bilayer. Then the authors used molecular dynamics simulation and demonstrated that a local high concentration of Ca²⁺ induced an ordered state of lipids. The experiments were well-designed and well-conducted. Data

were presented properly and seem highly intriguing in terms of the function of A5 itself. On the other hand, only a weak evidence has been shown on the “phase transition” and “stabilization” of lipids by A5, which the authors claimed is the new finding in this study. Especially, the interpretation of Figure 3, which is the most important figures in the manuscript, seems unclear to reach the conclusion that A5 stabilizes the lipid bilayer by inducing a phase transition. They repeatedly claimed that the formation of 2D-lattice of A5 induced a phase transition of lipids and restricted the mobility. However, their HS-AFM observation demonstrated that the height of the lipid bilayer had changed before A5 formed the 2D-lattice. In addition, there is no direct evidence that the lipid phase changed from liquid crystal to gel states. Since A5 strongly binds to lipids in a Ca²⁺-dependent manner, the lateral mobility of the lipids in the bilayer is restricted by 2D-lattice formation of A5. However, this is just a “restriction” of lipid movement by A5, but not a “phase transition” of lipids. The followings are the comments related to the issues mentioned above.

Major comments:

1. In Figure 3c and d, the authors demonstrated that the addition of A5 to the lipid bilayer shrank a patch of lipid bilayer on a flat substrate. Based on the time-course observation of the membrane area and A5 binding, they found 5 distinct periods in the A5 assembly process. Immediately after the addition of A5, a patch of lipid bilayer started to shrink (Figure 3c), resulting in a slow decrease of lipid area (Figure 3d). The authors claim that the shrinkage is caused by lipid-bound A5 molecules, although they are invisible on the AFM images due to their fast 2D-diffusion. This might be a possible interpretation, but other interpretations might also be possible, and even more feasible. For example, free A5 in the chamber deprived lipids from the edge of the 2D-lipid patch, or just accelerated the dissociation of lipids from the 2D-patch by unknown mechanisms. Therefore, the authors have to show a direct evidence of membrane “shrinkage”, to distinguish from a reduction of lipid molecules from the 2D-lipid bilayer.

Author Reply: To address reviewer’s comments, we have performed additional CLSM fluorescence microscopy experiments using GUVs (of the same lipid composition as in the HS-AFM experiment) to observe the impact of A5 self-assembly on the lipid phase (new Figure 5, page 10). While the A5 molecules self-assembled into 2D-lattice and fully covered on the GUV, the Laurdan fluorescence probe reported lipid order change, and also a shrinkage of the diameter of the GUV measured by confocal fluorescence image (new Figure 5e). Simultaneously, the lipid order determined by the membrane-incorporated dye, Laurdan, shifted to a higher value (new Figure 5g) reporting a phase transition in the membrane. While, we cannot exclude that some of the membrane shrinkage in the HS-AFM experiment is due to lipid shedding by an unknown mechanism, the fact that both techniques observe shrinkage concomitant to thickening (HS-AFM) and phase transition (CLSM), is strong evidence that the shrinkage is indeed linked with a change in lipid order, which is after all also what is expected to happen.

2. Following the membrane shrinkage in Period (1), the authors claimed that the lipid-bound A5 became visible as a small seeding/aggregate (Period (2)). However, it is very hard to see A5 molecules in Figure 3c and Movie S4. The authors should show the image and indicate the small patch of A5 by an arrow.

Author Reply: We indicate the small A5 aggregates by yellow arrows in Figure 3c.

3. In Figure 3e, the height of the lipid bilayer and that of A5 lattice from the mica surface during the A5 assembly was measured. At 49 seconds, when Period (1) ended and Period (2) started, the height of the membrane suddenly increased from 3.15 nm to 3.35 nm by 0.2 nm. In addition, the height of A5 was measured between 5.5 and 6.0 nm. However, based on Figure 3d, there is no A5 2D-lattice on the lipid bilayer during Period (2), although a small aggregate emerged in the beginning of Period (2) and immediately disappeared after several seconds. So, I wonder how they measured the height of A5 in the absence of A5 2D-lattice on the lipid bilayer. Does it mean the height of diffusive A5 on the lipid bilayer?

If so, the authors should show the A5 height plot in Period (1) as well. It is strange that Figure 3d does not contain the A5 height in Period (1).

Author Reply: We delineated and measured the number and characteristics of A5 aggregates using the function “Analyzed Particles” with a minimum size of 200 nm² in ImageJ. Thus, small A5 aggregates with size smaller than this minimum value are not counted in Figure 3d. The height measured in Figure 3e is based on the pixel height, this without any size constrains. To clarify this, we added the information concerning the size criteria into the legend of Figure 3d.

4. Related to the previous comment, the authors claimed that the sudden change in the height of the lipid bilayer in Period (2) was due to phase transition of lipids from liquid crystal to solid states. However, in this period (Period (2)), most of A5 molecules existed on the surface of the lipid bilayer in a dispersed and diffusive state (line 179-181), and had not formed 2D-lattice yet. This means that diffusive A5 caused a phase transition of the lipids. I wonder how this happened. The author should explain how phase transition had been caused by diffusive A5.

Author Reply: We expect that the diffusive A5 molecules readily influence the order of the underlying lipids – we have to remind that (i) even isolated A5 diffuse almost an order of magnitude slower than lipids in a bilayer(Heath G. R. & Scheuring S., Nat Commun, 2018, DOI: [10.1038/s41467-018-07512-3](https://doi.org/10.1038/s41467-018-07512-3)), and (ii) each A5 trimer ‘binds’ between 9 and 30 negatively charged lipids, and thus locally readily create order in the bilayer (without forming a 2D lattice). Our new CLSM experiments corroborate this finding.

5. In addition, the height of the lipid bilayer further increased in Period (3) by 0.1 nm. However, the authors did not explain how this happened. Is this another phase transition of lipids? The authors should clarify this.

Author Reply: No, we do not claim that this is another phase transition. Likely the lesser value in Period (2) is an average where some areas of the bilayer are still thinner. To address the reviewer’s comment, we performed additional experiments to study the lipid order using the membrane-incorporated dye, Laurdan (new Figure 5). In the time-lapse experiments of A5 additions (new Figure 5e-5g), we found that small and mobile A5 aggregates on GUVs can readily increase the average lipid order, and the formation of A5 2D lattices can further make the underlying lipids become overall more ordered. However, these details require further examination that is out of the scope for this study.

6. The phase transition occurred after the shrinkage of the lipid patch (Period (1)) stopped. It was not clearly stated what is the relationship between the membrane shrinkage and the phase transition of the lipids. The authors should clarify the relationship among the following issues: membrane shrinkage by diffusive A5 (Period (1)), phase transition of the lipids by diffusive A5 (Period (2)), further phase transition of the lipid (?) and 2D-assembly of A5 lattice (Period (3)).

Author Reply: For AFM-based characterization, the phase transition of lipids can be well-characterized by the height change (because AFM is particularly sensitive for measurements in the z-dimension) rather than membrane shrinkage. Thus, the height change should be a solid indicator for phase transition by AFM. Our new FRAP and CLSM data together with the AFM and the MDS data give now a clear picture how early adsorption of A5 readily leads to slowed diffusion and some (maybe local) ordering in the membrane, while the formation of the full 2D lattice leads to a dramatic slowdown of lipid diffusivity and order change of the C-tails of the lipids. Thickening and order change lead logically to some lateral shrinkage of the membrane.

7. In Figure 4, the authors performed MD simulation to mimic the membrane shrinkage and ordered state of lipids in the presence of high concentration of Ca²⁺. They claimed that lipid bilayer of DOPC/DOPS was shrunk in x-y direction and transformed into more ordered state in a Ca²⁺-dependent manner. They estimated a local concentration of Ca²⁺ induced by 2D-lattice of A5, and performed the simulation. However, based on Figure 3, the membrane shrinkage occurred in the absence of A5 2D-lattice (Period (1)), and stopped during the lattice formation (Periods (2) and (3)). Therefore, if they wanted to simulate the membrane shrinkage, they had to estimate local concentration of Ca²⁺ in Period (1), where A5 is dispersed and diffusive, and not visible by AFM.

Author Reply: We first used the crystallographic positions in the A5 2D lattice to confine Ca²⁺ ions in the MDS. Then rather high Ca²⁺ concentration (~500mM) were used in the simulation box to observe shrinkage (Fig. 6b) and order (Fig. 6c). As we detail along with Supplementary Information 1), under the assumption that Ca²⁺-ions are sandwiched in a layer of ~1 nm thickness, the local Ca²⁺-concentration near the membrane under a A5 lattice can be estimated between 110 mM (based on PDB 1BC0 with 2 Ca²⁺-ions) and 606 mM (based on PDB 2H0K with 11 Ca²⁺-ions), respectively. Thus, the MDS are performed under the conditions corresponding to the high estimate of the 2D-lattice. This is done, because (despite the tremendous progress in computation) full atomic MDS are still restricted to the low μ s time range. It would be impossible to detect relevant changes in MDS under conditions that would need seconds in experiments. Anyway, MDS show that negatively charged lipids never diffuse away from the Ca²⁺ “anchors” proposed by the protein; there is now reason to assume that this would be different for a single A5 on the membrane.

Minor comments

Line 135: Figure 2f should be Figure 2g?

Author Reply: We corrected the wrong assignment.

Legend of Figure 3: period (ii) and (iii) should period (2) and (3)?

Author Reply: We corrected the typos.

Reviewers' comments:

Reviewer #1 (Remarks to the Author):

In this revised version, Lin and coworkers have addressed the comments raised in my previous review. Importantly, they have carried out additional experiments to obtain more (direct) evidence for the conclusion that A5 binding affects membrane lipid order. By FRAP and Laurdan fluorescence recording Lin et al now show that lipid diffusion is reduced and membrane order increased after A5 binding and self assembly on the membrane.

The manuscript has clearly benefitted from the inclusion of new experiments. However, a few (minor) points still require clarification/correction.

1. Addressing my previous point 3 (artificial membrane model) the authors argue that they have deliberately chosen a minimal membrane lipid system. This is in principle ok but their reasoning should be mentioned in the manuscript.
2. Page 7, bottom (FRAP experiments). Here the text reads that the authors have analyzed SLBs consisting of DOPC/DOPS/NBD-PC (19:20:1). Please explain the lipid composition in more detail. DOPC:DOPS at 1:1, ie 50:50 doped with NBD-PC?
4. Laurdan experiments. The authors should explain why they chose to use GUVs (and not SLBs) in these experiments.
5. Fig 5b. The A5-AF647 stain looks patchy to me. Shouldn't a uniform labeling all around the GUV be expected at this full coverage?
6. Fig 5e and page 9, bottom. Here a reduction in diameter of one GUV is shown that appears to occur upon full coverage A5 binding. The authors should show more examples and a statistic analysis of GUV diameters before and after A5 binding to conclude that A5 binding results in a compaction of the GUV (due to increase in order).
7. Fig 5g and page 9, bottom. The first addition of A5 leads to a saturation of appr 10% and an increase of the GP value from 0 to 0.07. Why does this increase only occurs after a delay of about 5 min? And how can it be explained - it is difficult to envisage that this increase only stems from the more ordered lipids underneath the small A5 lattices as these only cover appr 10%.
8. Discussion, page 14, bottom. The authors argue that in the course of cell membrane wounding Ca²⁺-buffering by A5 will silence scramblases. Is there evidence for this Ca²⁺-buffering? Otherwise this rather speculative argument should be omitted.

Reviewer #2 (Remarks to the Author):

The authors of the manuscript "Annexin-A5 Stabilizes Membrane Defects by Inducing Lipid Phase Transition" conducted several additional experiments on the phase transition of lipid bilayer by using lipid vesicles and fluorescence microscopy. However, several questions on the interpretation of AFM images shown in Figure 3 have not been solved completely.

1. In Figure 3c and d, the authors demonstrated that the addition of A5 to the lipid bilayer shrank a patch of lipid bilayer on a flat substrate. Based on the time-course observation of the membrane area and A5 binding, they found 5 distinct periods in the A5 assembly process. Immediately after the addition of A5, a patch of lipid bilayer started to shrink (Figure 3c), resulting in a slow decrease of lipid area (Figure 3d). The authors claim that the shrinkage is caused by lipid-bound A5 molecules, although they are invisible on the AFM images due to their fast 2D-diffusion. This might be a possible interpretation, but other interpretations might also be possible, and even more feasible. For example, free A5 in the chamber deprived lipids from the edge of the 2D-lipid patch, or just accelerated the dissociation of lipids from the 2D-patch by unknown mechanisms. Therefore, the authors have to show a direct evidence of membrane "shrinkage", to distinguish from a reduction of lipid molecules from the 2D-lipid bilayer.

Author Reply: To address reviewer's comments, we have performed additional CLSM fluorescence microscopy experiments using GUVs (of the same lipid composition as in the HS-AFM experiment) to observe the impact of A5 self-assembly on the lipid phase (new Figure 5, page 10). While the A5 molecules self-assembled into 2D-lattice and fully covered on the GUV, the Laurdan

fluorescence probe reported lipid order change, and also a shrinkage of the diameter of the GUV measured by confocal fluorescence image (new Figure 5e). Simultaneously, the lipid order determined by the membrane-incorporated dye, Laurdan, shifted to a higher value (new Figure 5g) reporting a phase transition in the membrane. While, we cannot exclude that some of the membrane shrinkage in the HSAFM experiment is due to lipid shedding by an unknown mechanism, the fact that both techniques observe shrinkage concomitant to thickening (HS-AFM) and phase transition (CLSM), is strong evidence that the shrinkage is indeed linked with a change in lipid order, which is after all also what is expected to happen.

6. The phase transition occurred after the shrinkage of the lipid patch (Period (1)) stopped. It was not clearly stated what is the relationship between the membrane shrinkage and the phase transition of the lipids. The authors should clarify the relationship among the following issues: membrane shrinkage by diffusive A5 (Period (1)), phase transition of the lipids by diffusive A5 (Period (2)), further phase transition of the lipid (?) and 2D-assembly of A5 lattice (Period (3)).
Author Reply: For AFM-based characterization, the phase transition of lipids can be well-characterized by the height change (because AFM is particularly sensitive for measurements in the z-dimension) rather than membrane shrinkage. Thus, the height change should be a solid indicator for phase transition by AFM. Our new FRAP and CLSM data together with the AFM and the MDS data give now a clear picture how early adsorption of A5 readily leads to slowed diffusion and some (maybe local) ordering in the membrane, while the formation of the full 2D lattice leads to a dramatic slowdown of lipid diffusivity and order change of the C-tails of the lipids. Thickening and order change lead logically to some lateral shrinkage of the membrane.

>The authors' replies to above two comments seem to partially clarify the points that I raised. It seems convincing that the assembly of A5 into a lattice induces lipid phase transition. However, what was concerned here is the relationship between membrane shrinkage and the lipid phase transition that the authors observed in Figure 3c-e. The "membrane shrinkage" occurred in Period (1), and the phase transition occurred in Period (2). According to the authors' explanation "the phase transition of lipids can be well-characterized by the height change rather than membrane shrinkage", the phase transition occurred at the beginning of Period 2, which I completely agree. Then, what is the membrane shrinkage observed in Period (1)? Are they two independent events, or are there are causal relationships?

What is more confusing is the interpretation of the additional experiments using GUV and fluorescence dye which is sensitive to the lipid order. This experiment itself gives an additional evidence for the lipid phase transition induced by A5. However, it did not provide any additional information on the relationship between the membrane shrinkage and the phase transition, since they did not conducted time lapse analysis of the vesicle size, which indicates the process of the membrane shrinkage, due to a drifting of the focus in z direction. They just measured the vesicle size at the end of the observation. Therefore, it is not clear whether membrane shrinkage causes the lipid phase transition or not. The sentence "the fact that both techniques observe shrinkage concomitant to thickening (HS-AFM) and phase transition (CLSM), is strong evidence that the shrinkage is indeed linked with a change in lipid order, which is after all also what is expected to happen" is not convincing until they examine in the GUV experiment the time-course relationship between the vesicle size, which indicates the membrane shrinkage, and the Laurdan signal, which indicates the lipid order.

> In the additional experiments shown in Figure 5, the authors performed fluorescence-based analysis of lipid order and A5 binding. They claimed that the GP value significantly increased 280 sec after the first addition of A5, which covered 10 % of the GUV surface. When we compare this results and the AFM observation shown in Figure 3, I wonder why this large delay (280 sec) occurred. The authors did not give any explanations on this issue. In the AFM observation, the phase transition occurred at the beginning of Period (2), when the surface of the lipid bilayer has still not been covered by (diffusive) A5. Although they did not show any accurate quantification of A5-bound surface area in figure 2c (this point is also related to my next comment), it is apparently less than 10 % at 52 sec. So, authors should explain about this large discrepancy in the time course of phase transition between two different experimental systems.

3. In Figure 3e, the height of the lipid bilayer and that of A5 lattice from the mica surface during the A5 assembly was measured. At 49 seconds, when Period (1) ended and Period (2) started, the height of the membrane suddenly increased from 3.15 nm to 3.35 nm by 0.2 nm. In addition, the height of A5 was measured between 5.5 and 6.0 nm. However, based on Figure 3d, there is no A5 2D-lattice on the lipid bilayer during Period (2), although a small aggregate emerged in the beginning of Period (2) and immediately disappeared after several seconds. So, I wonder how they measured the height of A5 in the absence of A5 2D-lattice on the lipid bilayer. Does it mean the height of diffusive A5 on the lipid bilayer? If so, the authors should show the A5 height plot in Period (1) as well. It is strange that Figure 3d does not contain the A5 height in Period (1).
Author Reply: We delineated and measured the number and characteristics of A5 aggregates using the function "Analyzed Particles" with a minimum size of 200 nm² in ImageJ. Thus, small A5 aggregates with size smaller than this minimum value are not counted in Figure 3d. The height measured in Figure 3e is based on the pixel height, this without any size constraints. To clarify this, we added the information concerning the size criteria into the legend of Figure 3d.

> The authors' reply clarified the procedure of image analysis for extracting A5 aggregates. However, I still wonder how they measured "membrane height" together with "A5 height" from the same image. To do that, they have to separate A5-bound area from A5-free area. I guess they set ROIs (regions of interest) on the AFM image and separate them, but I wonder how they set the threshold for the ROI definition. They should show as a supplementary figure how they extract A5-bound area and A5-free area, and subsequently extract the area of A5 aggregation (by 200 nm² threshold).

Responses to Reviewer's Comments

Reviewer's comments: Black

Author's Response: Blue

Reviewers' comments:

Reviewer #1 (Remarks to the Author):

In this revised version, Lin and coworkers have addressed the comments raised in my previous review. Importantly, they have carried out additional experiments to obtain more (direct) evidence for the conclusion that A5 binding affects membrane lipid order. By FRAP and Laurdan fluorescence recording Lin et al now show that lipid diffusion is reduced and membrane order increased after A5 binding and self assembly on the membrane.

The manuscript has clearly benefitted from the inclusion of new experiments. However, a few (minor) points still require clarification/correction.

1. Addressing my previous point 3 (artificial membrane model) the authors argue that they have deliberately chosen a minimal membrane lipid system. This is in principle ok but their reasoning should be mentioned in the manuscript.

Author Reply: We address the reviewer's comment in the revised manuscript.

In Page 5: "This behavior was expected because the lipids in our minimal membrane system, dioleoylphosphatidylcholine/dioleoylphosphatidyl-serine (DOPC/DOPS) (50:50), have phase-transition temperatures of -17°C and -11°C, respectively, and, thus, form a fluid phase SLB at room temperature. Subsequent injection of A5 into the HS-AFM fluid cell elicited the study of the adsorption and self-assembly of A5 on these membrane patches in real time – this is likely also the case for other negatively charged lipids that interact with the A5 over Ca²⁺ or more complex lipid systems containing PS."

2. Page 7, bottom (FRAP experiments). Here the text reads that the authors have analyzed SLBs consisting of DOPC/DOPS/NBD-PC (19:20:1). Please explain the lipid composition in more detail. DOPC:DOPS at 1:1, ie 50:50 doped with NBD-PC?

Author Reply: Yes, this is correct, but instead of saying "doped with", we report the exact lipid composition of SLBs prepared for FRAP experiments, which is DOPC/DOPS/NBD-PC (19:20:1), *i.e.* the NBD-PC is 1/40 (2.5%). We clarify this on top of page 8.

4. Laurdan experiments. The authors should explain why they chose to use GUVs (and not SLBs) in these experiments.

Author Reply: We explain the reason for choosing GUVs instead of SLBs for Laurdan experiments on Page 9: "The usage of GUVs instead of SLBs was because of the relative orientation of the LAURDAN electronic transition moment with respect to the polarization plane of the excitation laser in CLSM, *i.e.* watching slices of membrane from the side at approximately the equatorial plane of the GUVs.³⁹ Thus, we employed CLSM focusing at the equatorial plane of free-standing GUVs doped with Laurdan (minimizing the photoselection effect³⁹), and recorded two spectral channels (channel 1: 410-463 nm

(*ch1*) and channel 2: 470-535 nm (*ch2*)), in A5-free (**Figure 5a**) and A5-full-coverage (**Figure 5b**) conditions, respectively.”

5. Fig 5b. The A5-AF647 stain looks patchy to me. Shouldn't a uniform labeling all around the GUV be expected at this full coverage?

Author Reply: As we noted in the manuscript, the relative focal plane change with respect to the GUV's equatorial section also changes when the diameter of the GUV varies. Those patchy feature can be attributed from a little z-drift of the equatorial section of GUV. The A5-AF647 stain appears also somewhat patchy in 5b, because the images shown are raw data images recorded in fast CLSM scanning mode and without averaging.

6. Fig 5e and page 9, bottom. Here a reduction in diameter of one GUV is shown that appears to occur upon full coverage A5 binding. The authors should show more examples and a statistic analysis of GUV diameters before and after A5 binding to conclude that A5 binding results in a compaction of the GUV (due to increase in order).

Author Reply: In response, we show now several GUV examples and their statistical analysis in a novel Supplementary Figure 3. We also note that most free-standing GUVs ruptured or moved away from the monitored region during the additions of A5 due to flow. Thus, only the GUVs that were clearly identified before and after A5-addition are reported here.

Figure S3. (a) Representative CLSM fluorescence images showing the impact of A5 2D-lattice self-assembly on GUVs. (b) The correlation of GUV diameter before and after A5 addition. (c) The correlation between diameter change and GP value change induced by the A5 2D-lattice self-assembly on GUVs. We note that most free-standing GUVs ruptured or moved away from the monitored region during the additions of A5, due to flow. Thus, only the GUVs that were clearly identified before and after A5-addition are reported here. When GUVs aggregated, we used partial, non-overlapping regions of the GUVs to calculate the mean GP value.

7. Fig 5g and page 9, bottom. The first addition of A5 leads to a saturation of appr 10% and an increase of the GP value from 0 to 0.07. Why does this increase only occurs after a delay of about 5 min? And how can it be explained - it is difficult to envisage that this increase only stems from the more ordered lipids underneath the small A5 lattices as these only cover appr 10%.

Author Reply: Our wording “partial A5-addition a significant increase of the GP value from ~0 to ~0.07 over a ~280s delay occurred and plateaued after 395.2s” was clearly not ideal. What we meant was that there was in 5g an increase of the GP value during the first ~280s of partial A5 coverage. We rephrase the sentence and clarify further, to: “In this time-lapse experiment, we observed that following the first, partial A5-addition a significant increase of the GP value from ~0 to ~0.07 occurred over the first ~280s and plateaued after 395.2s (**Figure 5g**). This observation combined with the A5 fluorescence channel provides further evidence that small, mobile A5 aggregates on GUVs can readily modulate the lipid order. The huge surface area of GUVs, ~110 μm^2 , much larger compared to the membrane patches monitored by HS-AFM (~0.1 μm^2 , **Figure 3c**) at similar diffusion speed of A5, could explain why it takes longer to observe an order change in the underlying lipids at low A5 surface coverage.”

8. Discussion, page 14, bottom. The authors argue that in the course of cell membrane wounding Ca²⁺-buffering by A5 will silence scramblases. Is there evidence for this Ca²⁺-buffering? Otherwise this rather speculative argument should be omitted.

Author Reply: We add a reference to indicate the capability of Ca²⁺-buffering by A5.

Reviewer #2 (Remarks to the Author):

The authors of the manuscript "Annexin-A5 Stabilizes Membrane Defects by Inducing Lipid Phase Transition" conducted several additional experiments on the phase transition of lipid bilayer by using lipid vesicles and fluorescence microscopy. However, several questions on the interpretation of AFM images shown in Figure 3 have not been solved completely.

1. In Figure 3c and d, the authors demonstrated that the addition of A5 to the lipid bilayer shrank a patch of lipid bilayer on a flat substrate. Based on the time-course observation of the membrane area and A5 binding, they found 5 distinct periods in the A5 assembly process. Immediately after the addition of A5, a patch of lipid bilayer started to shrink (Figure 3c), resulting in a slow decrease of lipid area (Figure 3d). The authors claim that the shrinkage is caused by lipid-bound A5 molecules, although they are invisible on the AFM images due to their fast 2D-diffusion. This might be a possible interpretation, but other interpretations might also be possible, and even more feasible. For example, free A5 in the chamber deprived lipids from the edge of the 2D-lipid patch, or just accelerated the dissociation of lipids from the 2D-patch by unknown mechanisms. Therefore, the authors have to show a direct evidence of membrane “shrinkage”, to distinguish from a reduction of lipid molecules from the 2D-lipid bilayer.

Author Reply: To address reviewer’s comments, we have performed additional CLSM fluorescence microscopy experiments using GUVs (of the same lipid composition as in the HS-AFM experiment) to observe the impact of A5 self-assembly on the lipid phase (new Figure 5, page 10). While the A5 molecules self-assembled into 2D-lattice and fully covered on the GUV, the Laurdan fluorescence probe reported lipid order change, and also a shrinkage of the diameter of the GUV measured by confocal fluorescence image (new Figure 5e). Simultaneously, the lipid order determined by the membrane-incorporated dye, Laurdan, shifted to a higher value (new Figure 5g) reporting a phase transition in the

membrane. While, we cannot exclude that some of the membrane shrinkage in the HSAFM experiment is due to lipid shedding by an unknown mechanism, the fact that both techniques observe shrinkage concomitant to thickening (HS-AFM) and phase transition (CLSM), is strong evidence that the shrinkage is indeed linked with a change in lipid order, which is after all also what is expected to happen.

6. The phase transition occurred after the shrinkage of the lipid patch (Period (1)) stopped. It was not clearly stated what is the relationship between the membrane shrinkage and the phase transition of the lipids. The authors should clarify the relationship among the following issues: membrane shrinkage by diffusive A5 (Period (1)), phase transition of the lipids by diffusive A5 (Period (2)), further phase transition of the lipid (?) and 2D-assembly of A5 lattice (Period (3)).

Author Reply: For AFM-based characterization, the phase transition of lipids can be well-characterized by the height change (because AFM is particularly sensitive for measurements in the z-dimension) rather than membrane shrinkage. Thus, the height change should be a solid indicator for phase transition by AFM. Our new FRAP and CLSM data together with the AFM and the MDS data give now a clear picture how early adsorption of A5 readily leads to slowed diffusion and some (maybe local) ordering in the membrane, while the formation of the full 2D lattice leads to a dramatic slowdown of lipid diffusivity and order change of the C-tails of the lipids. Thickening and order change lead logically to some lateral shrinkage of the membrane.

>The authors' replies to above two comments seem to partially clarify the points that I raised. It seems convincing that the assembly of A5 into a lattice induces lipid phase transition. However, what was concerned here is the relationship between membrane shrinkage and the lipid phase transition that the authors observed in Figure 3c-e. The "membrane shrinkage" occurred in Period (1), and the phase transition occurred in Period (2). According to the authors' explanation "the phase transition of lipids can be well-characterized by the height change rather than membrane shrinkage", the phase transition occurred at the beginning of Period 2, which I completely agree. Then, what is the membrane shrinkage observed in Period (1)? Are they two independent events, or are there are causal relationships? What is more confusing is the interpretation of the additional experiments using GUV and fluorescence dye which is sensitive to the lipid order. This experiment itself gives an additional evidence for the lipid phase transition induced by A5. However, it did not provide any additional information on the relationship between the membrane shrinkage and the phase transition, since they did not conducted time lapse analysis of the vesicle size, which indicates the process of the membrane shrinkage, due to a drifting of the focus in z direction. They just measured the vesicle size at the end of the observation. Therefore, it is not clear whether membrane shrinkage causes the lipid phase transition or not. The sentence "the fact that both techniques observe shrinkage concomitant to thickening (HS-AFM) and phase transition (CLSM), is strong evidence that the shrinkage is indeed linked with a change in lipid order, which is after all also what is expected to happen" is not convincing until they examine in the GUV experiment the time-course relationship between the vesicle size, which indicates the membrane shrinkage, and the Lauden signal, which indicates the lipid order.

Author Reply: We are sure that the membrane patches only shrink after supplementing A5 molecules into HS-AFM fluid cell, without A5 they are diffusive and of constant size on mica substrate as shown in Figure 3b. Thus, we believe the membrane shrinkage must be induced by impacts of A5. As suggested by the reviewer, we performed time lapse analysis of the vesicle size on the GUVs to observe "whether membrane shrinkage causes the lipid phase transition or not". The results are shown in the new Figure 5g (green trace), where two subsequent additions of A5 simultaneously induce the reduction in GUV's diameter and a change in the GP value. Especially, the second addition of A5 gives a strong time-dependent relationship between the vesicle size and the GP value (lipid order). Although, after the first

A5 addition, we only observed minor increase of GP together with a significant GUV size reduction. This could be explained by the continuous surface (and closed buffer-filled topology) of the GUV. In contrast, the membrane patch with finite size has different (open) membrane boundaries that can offer a more efficient way for membrane shrinkage during the phase transition of lipids. We have added in the first revision the comment “While the reason for this shrinkage remains unclear, and lipid shedding by an unknown A5-dependent mechanism cannot be excluded...” (line 181, page 6) in response to the reviewer’s concern.

> In the additional experiments shown in Figure 5, the authors performed fluorescence-based analysis of lipid order and A5 binding. They claimed that the GP value significantly increased 280 sec after the first addition of A5, which covered 10 % of the GUV surface. When we compare this results and the AFM observation shown in Figure 3, I wonder why this large delay (280 sec) occurred. The authors did not give any explanations on this issue. In the AFM observation, the phase transition occurred at the beginning of Period (2), when the surface of the lipid bilayer has still not been covered by (diffusive) A5. Although they did not show any accurate quantification of A5-bound surface area in figure 2c (this point is also related to my next comment), it is apparently less than 10 % at 52 sec. So, authors should explain about this large discrepancy in the time course of phase transition between two different experimental systems.

Author Reply: Our wording “partial A5-addition a significant increase of the GP value from ~0 to ~0.07 over a ~280s delay occurred and plateaued after 395.2s” was clearly not ideal. What we meant was that there was in 5g an increase of the GP value during the first ~280s of partial A5 coverage. We rephrase the sentence and clarify further, to: “In this time-lapse experiment, we observed that following the first, partial A5-addition a significant increase of the GP value from ~0 to ~0.07 occurred over the first ~280s and plateaued after 395.2s (**Figure 5g**). This observation combined with the A5 fluorescence channel provides further evidence that small, mobile A5 aggregates on GUVs can readily modulate the lipid order. The huge surface area of GUVs, ~110 μm^2 , much larger compared to the membrane patches monitored by HS-AFM (~0.1 μm^2 , **Figure 3c**) at similar diffusion speed of A5, could explain why it takes longer to observe an order change in the underlying lipids at low A5 surface coverage.”

We also explain the time course difference between HS-AFM and CLSM experiments on Page 10: “The time course between the HS-AFM and the CLSM experiments are somewhat different, because (i) the A5 assembly conditions are different, *i.e.* continuous in HS-AFM and step-wise in CLSM and (ii) the surface area of membrane patches (~0.1 μm^2) and GUVs (100 μm^2) are very different.”

3. In Figure 3e, the height of the lipid bilayer and that of A5 lattice from the mica surface during the A5 assembly was measured. At 49 seconds, when Period (1) ended and Period (2) started, the height of the membrane suddenly increased from 3.15 nm to 3.35 nm by 0.2 nm. In addition, the height of A5 was measured between 5.5 and 6.0 nm. However, based on Figure 3d, there is no A5 2D-lattice on the lipid bilayer during Period (2), although a small aggregate emerged in the beginning of Period (2) and immediately disappeared after several seconds. So, I wonder how they measured the height of A5 in the absence of A5 2D-lattice on the lipid bilayer. Does it mean the height of diffusive A5 on the lipid bilayer? If so, the authors should show the A5 height plot in Period (1) as well. It is strange that Figure 3d does not contain the A5 height in Period (1).

Author Reply: We delineated and measured the number and characteristics of A5 aggregates using the function “Analyzed Particles” with a minimum size of 200 nm² in ImageJ. Thus, small A5 aggregates with

size smaller than this minimum value are not counted in Figure 3d. The height measured in Figure 3e is based on the pixel height, this without any size constraints. To clarify this, we added the information concerning the size criteria into the legend of Figure 3d.

> The authors' reply clarified the procedure of image analysis for extracting A5 aggregates. However, I still wonder how they measured "membrane height" together with "A5 height" from the same image. To do that, they have to separate A5-bound area from A5-free area. I guess they set ROIs (regions of interest) on the AFM image and separate them, but I wonder how they set the threshold for the ROI definition. They should show as a supplementary figure how they extract A5-bound area and A5-free area, and subsequently extract the area of A5 aggregation (by 200 nm² threshold).

Author Reply: We have provided the detailed height histogram analysis in the supplementary information 2, and also described the procedures in the legend of Figure 2e. In general, the difference in height between the A5 molecules and the membrane is ~2.5 nm that enables us to determine their heights in the height histogram analyzed from the same image, and without the needs of threshold setting as mentioned by the reviewer.

REVIEWERS' COMMENTS:

Reviewer #1 (Remarks to the Author):

The authors have addressed my comments in an appropriate manner.

Reviewer #2 (Remarks to the Author):

In this revised version, the authors have addressed the comments that I raised in the previous review. They have carried out the experiment that I suggested and could successfully describe the relationship between membrane shrinkage and phase separation. In addition, they added more descriptions on the experimental procedures.